# Buffer layers for Test-Time Adaptation

**Hyeongyu Kim**[1]* **Geonhui Han**[1]* **Dosik Hwang**[1,2,3]†
[1]School of Electrical and Electronic Engineering, Yonsei University
[2]Department of Radiology and Center for Clinical Imaging Data Science,
College of Medicine, Yonsei University
[3]Artificial Intelligence and Robotics Institute, Korea Institute of Science and Technology
{lion4309, hgh6945, dosik.hwang@yonsei.ac.kr}

## Abstract

In recent advancements in Test Time Adaptation (TTA), most existing methodologies focus on updating normalization layers to adapt to the test domain. However, the reliance on normalization-based adaptation presents key challenges. First, normalization layers such as Batch Normalization (BN) are highly sensitive to small batch sizes, leading to unstable and inaccurate statistics. Moreover, normalization-based adaptation is inherently constrained by the structure of the pre-trained model, as it relies on training-time statistics that may not generalize well to unseen domains. These issues limit the effectiveness of normalization-based TTA approaches, especially under significant domain shift. In this paper, we introduce a novel paradigm based on the concept of a *Buffer* layer, which addresses the fundamental limitations of normalization layer updates. Unlike existing methods that modify the core parameters of the model, our approach preserves the integrity of the pre-trained backbone, inherently mitigating the risk of catastrophic forgetting during online adaptation. Through comprehensive experimentation, we demonstrate that our approach not only outperforms traditional methods in mitigating domain shift and enhancing model robustness, but also exhibits strong resilience to forgetting. Furthermore, our *Buffer* layer is modular and can be seamlessly integrated into nearly all existing TTA frameworks, resulting in consistent performance improvements across various architectures. These findings validate the effectiveness and versatility of the proposed solution in real-world domain adaptation scenarios. The code is available at `https://github.com/hyeongyu-kim/Buffer_TTA`.

## 1 Introduction

Recent progress in deep learning has led to remarkable advancements, largely driven by deep neural networks (DNNs) and large-scale datasets [2, 10], enabling unprecedented performance across various tasks. However, models still face challenges when tested on data that differs from the training data, known as domain shift, which occurs when the data distribution in the target domain diverges from the source domain [8]. To tackle this, several strategies have emerged, including domain adaptation (DA) and domain generalization (DG) [3, 26, 30]. DA adapts models to target domains using labeled source data and unlabeled target data, but it requires access to both source and target domain data, which is often not feasible in real-world settings. Furthermore, DA methods are typically not online and cannot continuously adapt after deployment. DG, on the other hand, aims to generalize across domains without requiring target domain data during training, making it useful for unseen domains, but it struggles with capturing domain-specific patterns and often needs a diverse set of source domains, which may not always be available.

---

*Equal contribution
†Corresponding author

39th Conference on Neural Information Processing Systems (NeurIPS 2025).

Test-Time Adaptation (TTA) has emerged as a practical solution to address domain shift, allowing models to adapt to the target domain during the inference phase typically without requiring access to source domain data [25]. The core idea of TTA is to adjust the model based on the unlabeled target domain data while leveraging the pre-trained model, focusing on minimizing the discrepancy between the target and source domains, often by using criteria such as entropy minimization. TTA methods typically update either a small subset of parameters, such as normalization layers (particularly Batch Normalization, BN) [14, 23, 17, 29, 16, 4, 12, 27, 11, 18, 6], or the entire model [19, 24, 1]. However, these strategies face key challenges. Normalization-based methods suffer from unreliable statistics with small batch sizes, while full model updates can incur high memory and computational costs due to backpropagation through the entire network. These limitations hinder the practical deployment of TTA in resource-constrained scenarios.

In this work, we revisit test-time adaptation by introducing a lightweight and modular *Buffer* layer that can be seamlessly inserted into any pre-trained model, enabling efficient adaptation during inference without modifying the backbone parameters. Unlike prior methods that rely on updating normalization layers or fine-tuning the entire model, our approach delegates the adaptation task to an auxiliary network specifically designed to mitigate domain shift at test time. This design circumvents the instability of normalization-based methods under small batch sizes and avoids the computational burden of full backpropagation through the main network. Moreover, *Buffer* layer can optionally be co-trained with normalization parameters, offering a flexible and extensible mechanism for robust adaptation in diverse deployment scenarios.

We evaluated our method across several widely used TTA benchmarks, including CIFAR10-C, CIFAR-10-W, CIFAR100-C, and ImageNet-C, comparing it against state-of-the-art techniques spanning both conventional and temporally aware adaptation methods. Our approach is highly modular and method-agnostic, allowing seamless integration into diverse frameworks for a wide range of TTA tasks. Notably, when combined with normalization-based adaptation, which is used in most existing TTA methods, our *Buffer* layer consistently provided additional gains, highlighting its compatibility and effectiveness as a general enhancement module. In addition, *Buffer* layer serves as an isolated adaptation unit, effectively mitigating catastrophic forgetting by preserving the original model parameters, which is a common limitation in conventional TTA approaches.

**Contributions**

(1) A modular and scalable adaptation framework: We introduce a plug-and-play *Buffer* layer that can be seamlessly integrated into any pre-trained model architecture. Its modularity makes it applicable across a wide range of test-time adaptation scenarios without altering the original network.

(2) Consistent performance gains and broad compatibility: The proposed *Buffer* layer delivers substantial accuracy improvements across diverse TTA benchmarks. It also enhances performance of existing normalization-based methods when combined, demonstrating strong compatibility.

(3) Forgetting-resilient adaptation design: By isolating adaptation into a dedicated component, our approach mitigates catastrophic forgetting of pre-trained knowledge, a common and critical limitation in many online TTA techniques.

## 2 Background

### 2.1 Test-Time Adaptation

TTA has emerged as a practical solution to domain shift, enabling models to adjust to target distributions during inference without requiring access to source domain data. By leveraging a pre-trained model and adapting it on unlabeled target samples, TTA eliminates the need for retraining or supervision, making it appealing for real-world deployment under privacy and computational constraints. Existing TTA methods can be broadly categorized based on what components of the model are adapted and how the adaptation is performed. One common approach is BN adaptation, where methods such as [14, 23, 18, 17] update BN affine parameters via entropy minimization. Another category is pseudo-label-based adaptation, which uses confident model predictions as supervisory signals [27, 24, 7, 15, 1]. Methods like [28, 16, 21] leverage test-time augmentation and consistency regularization to improve robustness. Despite their differences, these approaches exhibit trade-offs between flexibility, stability, computational cost, and applicability in source-free settings.

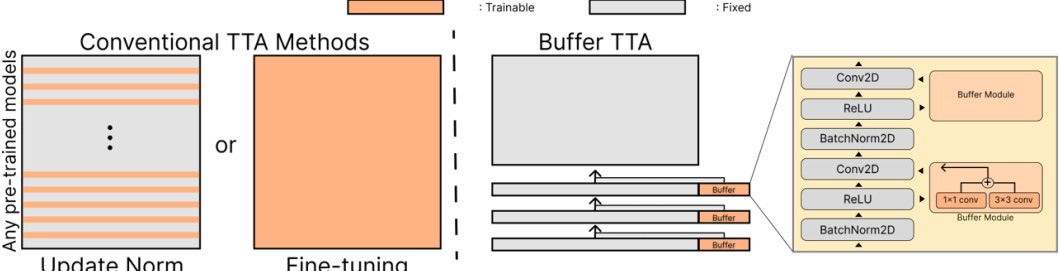

Figure 1: Overview of our test-time adaptation framework. Unlike prior methods that rely on updating normalization layers or fine-tuning the entire model, which require backpropagation and can suffer from instability under small batch sizes, or additive modules that require warm-up phases, our proposed *Buffer* layer enables direct test-time adaptation without any additional training. It operates on any type of objectives to mitigate domain shift, acting as a lightweight and modular adaptation unit that preserves the original model parameters and prevents catastrophic forgetting.

## 2.2 Limitations and Reformulation of the Problem

Despite these advancements, most TTA research has predominantly focused on ***how to update*** a model, for example through entropy minimization, consistency regularization, or confident sample selection, while paying relatively little attention to ***what to update***. Typical update targets include either the full model or BN layers, both of which involve trade-offs between stability and computational efficiency.

Although more recent works such as EcoTTA [21] and L-TTA [20] attempt to address these limitations by freezing the backbone and adapting only lightweight auxiliary components, they still require a *warm-up phase* using source data for initialization. This reliance on the source domain undermines the premise of source-free adaptation and limits their applicability in practical deployment scenarios.

In this work, we reformulate the problem by proposing to update a *dedicated auxiliary layer*, referred to as the *Buffer* layer, which is integrated into the network in a modular fashion. Unlike existing approaches, our method delegates the adaptation process to this compact and trainable unit, leaving the backbone untouched. This design mitigates catastrophic forgetting and supports stable adaptation under small batch sizes while maintaining computational efficiency. Moreover, our approach operates in a fully online and truly source-free manner, requiring **no additional initialization** or access to source data at any stage.

## 3  Methods

### 3.1  Motivation

TTA methods have predominantly focused on updating normalization layers, particularly batch normalization (BN), under the assumption that domain shift can be mitigated by adapting normalization statistics and affine parameters, i.e., $(\mu_s, \sigma_s, \gamma_s, \beta_s)$ and their target counterparts $(\mu_t, \sigma_t, \gamma_t, \beta_t)$. While intuitive, this assumption reduces domain shift to marginal distribution alignment and neglects more complex, class-conditional, or higher-order shifts.

Moreover, normalization layers are not explicitly designed for adaptation. Their reliance on batch statistics makes them unstable under small batch sizes, and modifying them risks disrupting the pretrained backbone. This motivates us to move beyond normalization layers and explore a structurally decoupled alternative.

In contrast to conventional BN-based methods, we propose a lightweight and modular *Buffer* layer that serves as an external adaptation unit, inserted into the network without altering the original architecture. By isolating adaptation using this auxiliary layer, we achieve a scalable and stable mechanism for source-free TTA, without relying on BN statistics or internal parameters of the model.

### 3.2 *Buffer* layer: Modular Adaptation for Source-Free TTA

We introduce the *Buffer* layer, a lightweight convolutional module for test-time adaptation, inserted in parallel to the pretrained backbone without altering its original parameters. As illustrated in Fig.1, each *Buffer* layer consists of simple 1×1 and 3×3 convolutions whose outputs are scaled by a learnable coefficient, and residually added to the original activations. *Buffer* layers are only inserted in the early stages of the network and can be optimized using any standard TTA objective, such as entropy minimization or consistency regularization.

In contrast to prior approaches that directly update BN layers, our method externalizes adaptation to a modular unit that only activates during inference. This ensures stable updates even under small batch sizes and avoids disrupting pretrained statistics. Furthermore, the *Buffer* layer can be optionally co-trained with normalization parameters, offering flexibility to adapt jointly when batch statistics are reliable. See Sec. 4.3.1. for architectural details.

## 4 Experiments

### 4.1 Baseline methods

We revisit a range of representative TTA methods and reinterpret them from the perspective of ***what*** and ***how*** to update. Many state-of-the-art approaches, such as TENT [23], CMF [11], EATA [17], SAR [18], DeYO [12], and ROID [16], are built upon updating BN layers in various ways. For example, TENT minimizes prediction entropy to update BN affine parameters, while EATA adds sample filtering and regularization strategies. Despite their differences in how they update parameters, these methods share a common design choice in what they update, namely the BN layers.

In our experiments, we retain each method's original update mechanism (***how*** to update) but replace the update target (***what*** to update) from BN (@BN) to our proposed *Buffer* layer (@*Buffer*). This setup enables a controlled comparison to isolate the effect of changing the adaptation unit itself. Specifically, we implement TENT@*Buffer*, EATA@*Buffer*, SAR@*Buffer*, and so forth, where only *Buffer* layers are updated during test time while the rest of the backbone including BN layers remain frozen. This design highlights the adaptability and generalizability of our *Buffer* layer across diverse TTA strategies. All experiments were conducted with three different random seeds, and the reported results represent either the mean or the mean with standard deviation.

### 4.2 Results

#### 4.2.1 CIFAR 10-C & CIFAR 100-C

Table 1: Classification error rate (%) on CIFAR10-C and CIFAR100-C. Batch sizes (BS) of 2, 4, and 16 are evaluated across WRN28, ResNeXT (CIFAR-10C) and WRN40, ResNeXT (CIFAR-100C).

| Method | | WRN28 (CIFAR10-C) | | | ResNeXT (CIFAR-10C) | | | WRN40 (CIFAR100-C) | | | ResNeXT (CIFAR100-C) | | |
|---|---|---|---|---|---|---|---|---|---|---|---|---|---|
| | | BS=2 | BS=4 | BS=16 | BS=2 | BS=4 | BS=16 | BS=2 | BS=4 | BS=16 | BS=2 | BS=4 | BS=16 |
| Source | | 43.52 | | | 17.98 | | | 46.75 | | | 46.44 | | |
| BN[14] | | $32.91_{\pm0.07}$ | $31.89_{\pm0.12}$ | $30.90_{\pm0.02}$ | $\mathbf{13.82}_{\pm0.08}$ | $\mathbf{13.26}_{\pm0.03}$ | $12.76_{\pm0.03}$ | $\mathbf{42.54}_{\pm0.09}$ | $\mathbf{41.79}_{\pm0.06}$ | $40.39_{\pm0.25}$ | $42.31_{\pm0.05}$ | $\mathbf{38.40}_{\pm0.05}$ | $37.30_{\pm0.06}$ |
| TENT[23] | @BN | $82.56_{\pm0.40}$ | $51.36_{\pm2.46}$ | $23.12_{\pm0.28}$ | $84.69_{\pm0.49}$ | $66.90_{\pm1.82}$ | $17.85_{\pm0.58}$ | $98.11_{\pm0.06}$ | $87.70_{\pm1.02}$ | $40.47_{\pm0.06}$ | $98.44_{\pm0.06}$ | $94.59_{\pm0.26}$ | $50.74_{\pm1.43}$ |
| | @Buffer | $37.05_{\pm0.31}$ (▼45.51) | $29.11_{\pm0.33}$ (▼22.25) | $20.32_{\pm0.13}$ (▼2.80) | $29.62_{\pm0.36}$ (▼55.07) | $19.35_{\pm0.23}$ (▼47.55) | $11.44_{\pm0.12}$ (▼6.40) | $89.90_{\pm0.16}$ (▼8.21) | $55.72_{\pm0.19}$ (▼31.98) | $39.30_{\pm0.06}$ (▼1.17) | $91.87_{\pm0.88}$ (▼6.57) | $48.17_{\pm0.15}$ (▼46.42) | $34.06_{\pm0.08}$ (▼16.68) |
| EATA[17] | @BN | $45.49_{\pm0.64}$ | $34.18_{\pm0.26}$ | $20.97_{\pm0.15}$ | $57.12_{\pm0.77}$ | $33.34_{\pm1.07}$ | $14.13_{\pm0.32}$ | $80.58_{\pm0.19}$ | $58.35_{\pm0.37}$ | $40.18_{\pm0.13}$ | $73.27_{\pm0.62}$ | $67.38_{\pm0.36}$ | $36.58_{\pm0.30}$ |
| | @Buffer | $35.41_{\pm0.34}$ (▼10.08) | $28.97_{\pm0.33}$ (▼5.21) | $19.91_{\pm0.06}$ (▼1.06) | $28.82_{\pm0.37}$ (▼28.32) | $19.37_{\pm0.16}$ (▼13.97) | $11.38_{\pm0.07}$ (▼2.75) | $80.43_{\pm0.13}$ (▼0.15) | $56.31_{\pm0.29}$ (▼2.04) | $39.94_{\pm0.25}$ (▼0.24) | $73.14_{\pm0.09}$ (▼0.13) | $48.17_{\pm0.15}$ (▼19.21) | $34.59_{\pm0.10}$ (▼1.99) |
| SAR[18] | @BN | $40.43_{\pm0.26}$ | $31.37_{\pm0.25}$ | $22.94_{\pm0.07}$ | $42.63_{\pm0.55}$ | $24.76_{\pm0.22}$ | $15.29_{\pm0.05}$ | $80.59_{\pm0.18}$ | $67.17_{\pm0.50}$ | $40.02_{\pm0.06}$ | $73.23_{\pm0.09}$ | $65.45_{\pm0.41}$ | $36.79_{\pm0.51}$ |
| | @Buffer | $38.27_{\pm0.11}$ (▼2.16) | $31.25_{\pm0.18}$ (▼0.12) | $23.06_{\pm0.03}$ (▲0.12) | $34.33_{\pm0.31}$ (▼8.30) | $24.39_{\pm0.14}$ (▼0.37) | $15.88_{\pm0.07}$ (▲0.59) | $80.56_{\pm0.33}$ (▼0.03) | $58.51_{\pm0.54}$ (▼8.66) | $43.67_{\pm0.35}$ (▲3.65) | $73.14_{\pm0.09}$ (▼0.09) | $52.99_{\pm0.03}$ (▼12.46) | $39.67_{\pm0.09}$ (▲2.88) |
| DeYo[12] | @BN | $69.95_{\pm0.91}$ | $37.65_{\pm1.05}$ | $21.96_{\pm0.05}$ | $76.53_{\pm0.53}$ | $36.71_{\pm0.67}$ | $14.11_{\pm0.20}$ | $80.83_{\pm0.23}$ | $76.87_{\pm1.85}$ | $39.76_{\pm0.08}$ | $73.33_{\pm0.11}$ | $92.18_{\pm0.36}$ | $38.23_{\pm0.62}$ |
| | @Buffer | $35.65_{\pm0.41}$ (▼34.30) | $28.97_{\pm0.35}$ (▼8.68) | $20.29_{\pm0.11}$ (▼1.67) | $28.61_{\pm0.36}$ (▼47.92) | $19.17_{\pm0.24}$ (▼17.54) | $11.40_{\pm0.03}$ (▼2.71) | $80.55_{\pm0.48}$ (▼0.28) | $56.23_{\pm0.15}$ (▼20.64) | $39.17_{\pm0.09}$ (▼0.59) | $73.15_{\pm0.09}$ (▼0.18) | $48.34_{\pm0.28}$ (▼43.84) | $33.84_{\pm0.14}$ (▼4.39) |
| CMF[11] | @BN | $41.42_{\pm0.73}$ | $28.51_{\pm0.26}$ | $19.06_{\pm0.12}$ | $53.45_{\pm1.12}$ | $21.31_{\pm0.20}$ | $11.71_{\pm0.04}$ | $95.59_{\pm0.17}$ | $56.09_{\pm0.26}$ | $39.19_{\pm0.05}$ | $98.07_{\pm0.07}$ | $74.29_{\pm2.41}$ | $34.02_{\pm0.24}$ |
| | @Buffer | $36.26_{\pm0.21}$ (▼5.16) | $27.91_{\pm0.21}$ (▼0.60) | $18.97_{\pm0.05}$ (▼0.09) | $29.83_{\pm2.26}$ (▼23.62) | $18.85_{\pm0.26}$ (▼2.46) | $\mathbf{10.88}_{\pm0.05}$ (▼0.83) | $81.04_{\pm0.61}$ (▼14.55) | $55.17_{\pm0.38}$ (▼0.92) | $\mathbf{39.13}_{\pm0.08}$ (▼0.06) | $71.90_{\pm0.29}$ (▼26.17) | $48.14_{\pm0.31}$ (▼26.15) | $\mathbf{33.61}_{\pm0.13}$ (▼0.41) |
| ROID[16] | @BN | $38.14_{\pm0.19}$ | $29.91_{\pm0.23}$ | $20.20_{\pm0.10}$ | $32.73_{\pm0.21}$ | $20.76_{\pm0.24}$ | $11.55_{\pm0.12}$ | $83.12_{\pm0.07}$ | $57.46_{\pm0.16}$ | $40.07_{\pm0.05}$ | $93.22_{\pm0.62}$ | $51.95_{\pm0.16}$ | $34.01_{\pm0.08}$ |
| | @Buffer | $38.06_{\pm0.11}$ (▼0.08) | $29.87_{\pm0.29}$ (▼0.04) | $20.09_{\pm0.19}$ (▼0.11) | $32.01_{\pm0.26}$ (▼0.72) | $20.67_{\pm0.17}$ (▼0.09) | $11.47_{\pm0.07}$ (▼0.08) | $80.77_{\pm0.28}$ (▼2.35) | $57.33_{\pm0.34}$ (▼0.13) | $40.65_{\pm0.45}$ (▲0.58) | $72.92_{\pm0.17}$ (▼20.30) | $50.67_{\pm0.26}$ (▼1.28) | $35.15_{\pm0.07}$ (▲1.14) |
| RoTTA[27] | @BN | $21.55_{\pm0.10}$ | $21.57_{\pm0.08}$ | $21.57_{\pm0.09}$ | $17.71_{\pm0.13}$ | $17.72_{\pm0.14}$ | $17.71_{\pm0.12}$ | $44.58_{\pm0.04}$ | $44.56_{\pm0.05}$ | $44.54_{\pm0.04}$ | $42.14_{\pm0.06}$ | $42.14_{\pm0.05}$ | $42.14_{\pm0.06}$ |
| | @Buffer | $21.27_{\pm0.11}$ (▼0.28) | $\mathbf{21.28}_{\pm0.11}$ (▼0.29) | $21.28_{\pm0.10}$ (▼0.29) | $17.52_{\pm0.11}$ (▼0.19) | $17.45_{\pm0.12}$ (▼0.27) | $17.38_{\pm0.12}$ (▼0.33) | $43.68_{\pm0.07}$ (▼0.98) | $42.55_{\pm0.06}$ (▼2.01) | $42.16_{\pm0.07}$ (▼2.38) | $\mathbf{41.58}_{\pm0.05}$ (▼0.56) | $41.50_{\pm0.07}$ (▼0.64) | $41.60_{\pm0.07}$ (▼0.54) |
| CoTTA[24] | | $70.54_{\pm0.88}$ | $48.59_{\pm0.65}$ | $22.58_{\pm0.23}$ | $63.06_{\pm0.72}$ | $30.53_{\pm0.35}$ | $13.76_{\pm0.21}$ | $90.84_{\pm0.55}$ | $63.00_{\pm0.47}$ | $43.18_{\pm0.32}$ | $91.16_{\pm0.65}$ | $62.97_{\pm0.35}$ | $38.33_{\pm0.17}$ |
| AdaContrast[1] | | $28.05_{\pm0.12}$ | $22.65_{\pm0.13}$ | $\mathbf{18.38}_{\pm0.01}$ | $38.19_{\pm0.96}$ | $21.20_{\pm0.15}$ | $11.39_{\pm0.03}$ | $63.96_{\pm0.36}$ | $51.96_{\pm0.33}$ | $39.72_{\pm0.17}$ | $67.44_{\pm0.20}$ | $53.01_{\pm0.18}$ | $36.75_{\pm0.15}$ |

Table 1 reports the classification error rates on CIFAR10-C and CIFAR100-C under various small test-time batch sizes (2, 4, and 16). Across all evaluated architectures, including WRN28, WRN40,

and ResNeXT, we observe that integrating the proposed *Buffer* layer (@*Buffer*) consistently and substantially improves performance over the original BN-based counterparts.

The performance gains are particularly notable in low batch-size regimes such as BS=2 and BS=4, where traditional BN-based adaptation methods often suffer due to unreliable batch statistics. For example, TENT @*Buffer* reduces error by up to 45.51% on WRN28 under BS=2, and similar trends are observed with DeYo. These results demonstrate that our *Buffer* layer not only generalizes well across diverse TTA algorithms but also offers a robust alternative to BN in small batch size scenarios.

### 4.2.2 CIFAR-10-W

Table 2. shows the results on CIFAR10-W, a challenging benchmark. Despite the difficulty, our *Buffer* layer consistently improves performance across all baselines. Notably, under small batch sizes (e.g., BS=2), our method achieves substantial gains: for instance, TENT from 89.30% to 64.14%, and DeYO from 84.37% to 46.96% when only the *Buffer* layer is updated. This demonstrates the effectiveness of our approach in overcoming the instability of normalization-based methods under limited statistical contexts.

However, as the batch size increases, the performance gap between @*Buffer* and the baseline narrows, and in some cases, even underperforms compared to the original method. Interestingly, in most cases, when both the *Buffer* layer and normalization parameters are updated jointly

Table 2: CIFAR10-W. **Bests are in bold.** Green background indicates performance improvement.

| Dataset | | CIFAR10W | | |
|---|---|---|---|---|
| Method | BS | 2 | 4 | 16 |
| Source | | 77.28 | | |
| TENT [23] | @BN | 89.30±0.43 | 84.72±1.09 | 59.66±2.65 |
| | @Buffer | 64.14±0.40 | 41.60±1.45 | 30.30±0.26 |
| | @BN+Buffer | 88.02±1.25 | 83.74±1.28 | 57.56±2.84 |
| EATA [17] | @BN | 68.94±1.06 | 59.59±2.01 | 39.71±0.86 |
| | @Buffer | **38.89**±0.12 | 36.02±0.42 | 29.98±0.17 |
| | @BN+Buffer | 70.91±0.45 | 59.53±1.01 | 38.96±0.60 |
| CMF [11] | @BN | 47.78±0.98 | 35.65±0.12 | 29.28±0.02 |
| | @Buffer | 41.27±0.33 | 36.00±0.14 | 30.63±0.24 |
| | @BN+Buffer | 62.90±2.42 | 35.43±0.02 | 29.22±0.12 |
| DeYo [12] | @BN | 84.37±0.61 | 63.22±0.67 | 39.60±0.53 |
| | @Buffer | 46.96±1.48 | **34.81**±0.26 | **28.30**±0.14 |
| | @BN+Buffer | 81.96±0.51 | 62.62±1.85 | 39.05±0.40 |
| SAR [18] | @BN | 42.26±0.09 | 36.95±0.02 | 30.73±0.02 |
| | @Buffer | 42.03±0.02 | 37.00±0.02 | 30.86±0.02 |
| | @BN+Buffer | 42.35±0.15 | 36.81±0.09 | 30.65±0.01 |
| ROID [16] | @BN | 41.81±0.09 | 36.24±0.04 | 29.93±0.03 |
| | @Buffer | 41.90±0.14 | 36.30±0.10 | 30.06±0.07 |
| | @BN+Buffer | 41.77±0.03 | 36.16±0.01 | 29.89±0.02 |

during adaptation (@BN+*Buffer*), the performance consistently improved. We attribute this to the varying optimization difficulty inherent in each method; methods like CMF and SAR may benefit more from additional degrees of freedom during adaptation. This suggests that the *Buffer* layer can still contribute positively when combined with normalization updates, even in cases where frozen BN alone is insufficient. Moreover, as batch size increases, the influence of BN becomes more prominent and reliable, making joint updates with the *Buffer* layer more beneficial. These observations highlight the effectiveness of the co-updating strategy in leveraging the strengths of both components.

### 4.2.3 Would *Buffer* work with GroupNorm? : ImageNet-C Results

Table 3: ImageNet-C. **Bests are in bold.** Green background indicates performance improvement.

| Dataset | | ImageNet-C | | | | | |
|---|---|---|---|---|---|---|---|
| Models | | Res50(BN) | | | Resv2_50(GN) | | |
| Method \| BS | | 2 | 4 | 16 | 2 | 4 | 16 |
| Source | | 82.03 | | | 72.80 | | |
| TENT [23] | @BN | 96.29±0.11 | 78.90±0.16 | 63.61±0.21 | 94.51±0.07 | 85.63±0.07 | 72.81±0.21 |
| | @Buffer | 93.14±0.04 | 81.11±0.14 | 71.04±0.01 | 94.21±0.06 | 85.43±0.10 | 72.24±0.18 |
| | @BN+Buffer | 96.15±0.13 | 78.66±0.13 | 63.48±0.15 | 94.55±0.07 | 85.49±0.12 | 72.30±0.33 |
| EATA [17] | @BN | 93.30±0.04 | 80.32±0.02 | 62.88±0.82 | 94.41±0.08 | 85.64±0.08 | 72.32±0.15 |
| | @Buffer | 93.22±0.04 | 81.12±0.10 | 71.09±0.04 | 94.25±0.08 | 91.43±0.34 | 71.95±0.13 |
| | @BN+Buffer | 93.28±0.04 | 80.28±0.05 | 62.16±0.11 | 94.44±0.03 | 85.43±0.34 | 70.83±0.15 |
| CMF [11] | @BN | 99.32±0.01 | 97.56±0.29 | 64.80±0.04 | 96.72±0.25 | 83.23±0.11 | 69.30±0.05 |
| | @Buffer | 93.24±0.03 | 80.95±0.06 | 71.03±0.14 | 95.28±0.11 | 89.89±0.27 | 69.52±0.41 |
| | @BN+Buffer | 99.35±0.06 | 97.89±0.31 | 64.46±0.05 | 97.45±0.07 | **82.01**±0.40 | **65.51**±0.03 |
| DeYo [12] | @BN | 93.29±0.04 | 81.51±0.29 | 64.86±0.74 | 94.44±0.08 | 85.55±0.10 | 70.98±0.15 |
| | @Buffer | 93.23±0.03 | 81.11±0.08 | 70.81±0.22 | 94.24±0.07 | 85.57±0.17 | 73.84±0.82 |
| | @BN+Buffer | 93.29±0.03 | 90.81±0.85 | 68.48±0.45 | 94.25±0.08 | 85.70±0.17 | 69.16±0.51 |
| SAR [18] | @BN | 93.31±0.04 | 81.07±0.10 | 66.59±0.32 | 94.34±0.05 | 85.67±0.12 | 72.91±0.25 |
| | @Buffer | 93.25±0.04 | 81.03±0.06 | 71.09±0.05 | 94.25±0.06 | 85.45±0.13 | 72.60±0.09 |
| | @BN+Buffer | 93.32±0.04 | 81.05±0.31 | 65.53±0.31 | 94.46±0.04 | 85.45±0.16 | 72.69±0.25 |
| ROID [16] | @BN | 97.22±1.74 | 87.88±0.63 | 61.51±0.18 | 94.68±0.38 | 85.07±0.08 | 70.70±0.07 |
| | @Buffer | 93.27±0.03 | 81.18±0.09 | 70.56±0.15 | 94.32±0.05 | 86.48±0.09 | 69.59±0.23 |
| | @BN+Buffer | 97.12±0.12 | 87.69±0.12 | **61.35**±0.17 | 94.51±0.06 | 84.00±0.16 | 67.98±0.26 |

Similar to the observations on CIFAR-10-W, ImageNet-C results reveal that jointly updating the *Buffer* layer and norm layers becomes effective as the batch size grows (Table 3). This trend reinforces the notion that BN statistics become more stable and reliable with larger batches, allowing co-adaptation with the *Buffer* layer to yield improved performance. As previously noted in [18], however, BN causes instability during test-time adaptation, especially under small batch regimes.

To address this limitation and further demonstrate the model-agnostic nature of our *Buffer* layer, we extended our evaluation to a ResNetV2 [9] architecture that employs Group Normalization (GN) instead of BN. Experimental results on ImageNet-C with the GN-based ResNetV2 show trends consistent with those observed in the BN-based ResNet50, indicating that our *Buffer* layer remains effective regardless of the type of normalization used. These findings highlight the flexibility and general applicability of our method across diverse normalization schemes and network architectures.

### 4.2.4 On Large Batchsizes

Based on previous observations, we further evaluate the performance of our method under large-batch settings by jointly updating both the *Buffer* layer and BN parameters (Fig.2) . This setting aligns with the practical regime where BN benefits from stable batch-level statistics. Across datasets, including ImageNet-C, this co-adaptation strategy continues to yield improved performance, surpassing the baselines. These results confirm that the *Buffer* layer remains effective even when normalization-based adaptation is reliable, highlighting its complementary role in diverse test-time scenarios.

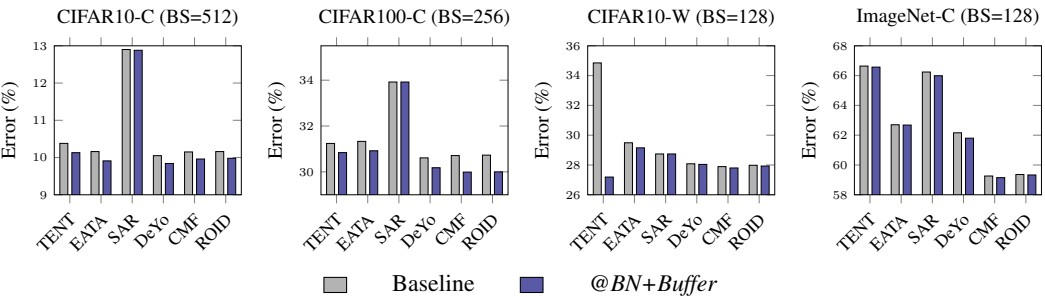

Figure 2: Classification error rates (%) across datasets on large batch sizes.

### 4.2.5 Continuously Changing Domains

Table 4: Classification error rate (%) on CIFAR10-C and CIFAR100-C under continuously changing environments, comparing models with and without *Buffer*. Accuracy is averaged over three different random seeds. **Bold** numbers indicate the highest accuracy.

| | Method | Gauss. | Shot | Implu. | Defoc. | Glass | Motio. | Zoom | Snow | Frost | Fog | Brigh. | Contr. | Elast. | Pixel | Jpeg | Avg |
|---|---|---|---|---|---|---|---|---|---|---|---|---|---|---|---|---|---|
| CIFAR10-C | TENT[23] @BN | 65.32 | 82.03 | 88.00 | 86.72 | 83.63 | 84.32 | 88.11 | 88.49 | 90.71 | 91.11 | 90.42 | 91.53 | 91.53 | 90.65 | 90.91 | 86.90 |
| | @*Buffer* | 35.30 | 31.42 | 39.06 | 24.99 | 44.43 | 25.06 | 24.48 | 26.66 | 26.46 | 23.84 | 19.94 | 22.45 | 33.47 | 27.57 | 32.51 | 29.17 |
| | EATA[17] @BN | 40.86 | 44.83 | 58.17 | 52.95 | 64.89 | 59.73 | 49.86 | 49.82 | 52.93 | 54.47 | 47.37 | 49.35 | 61.20 | 59.97 | 62.34 | 53.92 |
| | @*Buffer* | 34.50 | 30.20 | 37.54 | 24.97 | 42.66 | 25.29 | 24.11 | 27.04 | 26.05 | 24.33 | 19.52 | 21.38 | 33.49 | 27.38 | 32.41 | 28.72 |
| | SAR[18] @BN | 38.47 | 36.50 | 44.47 | 24.27 | 46.55 | 24.88 | 24.72 | 28.44 | 28.79 | 25.74 | 19.44 | 22.45 | 35.52 | 31.44 | 37.75 | 31.30 |
| | @*Buffer* | 39.00 | 37.06 | 45.57 | 24.31 | 46.82 | 24.90 | 24.64 | 28.77 | 29.06 | 25.86 | 19.56 | 22.57 | 35.24 | 31.99 | 38.75 | 31.61 |
| | DeYo[12] @BN | 41.37 | 49.16 | 66.30 | 66.54 | 75.58 | 71.28 | 71.96 | 72.95 | 71.85 | 74.82 | 70.80 | 71.47 | 74.91 | 76.71 | 77.75 | 68.90 |
| | @*Buffer* | 35.02 | 30.35 | 38.23 | 23.70 | 44.26 | 25.40 | 23.79 | 26.15 | 25.81 | 23.34 | 19.31 | 20.72 | 34.71 | 27.04 | 32.27 | 28.67 |
| | CMF[11] @BN | 34.12 | 31.48 | 39.06 | 22.82 | 41.70 | 23.36 | 22.44 | 26.08 | 25.88 | 23.35 | 18.22 | 19.49 | 33.59 | 27.70 | 33.74 | 28.20 |
| | @*Buffer* | 32.62 | 29.81 | 38.22 | 23.60 | 41.43 | 24.04 | 22.87 | 25.61 | 26.16 | 22.54 | 18.41 | 19.32 | 33.20 | 26.81 | 31.44 | **27.74** |
| | ROID[16] @BN | 36.81 | 35.19 | 42.90 | 23.53 | 44.66 | 23.64 | 23.42 | 27.26 | 27.55 | 24.28 | 18.19 | 20.40 | 34.68 | 29.80 | 36.95 | 29.95 |
| | @*Buffer* | 36.32 | 34.33 | 42.28 | 23.45 | 45.23 | 24.25 | 23.81 | 26.95 | 28.15 | 24.51 | 18.35 | 21.39 | 34.41 | 29.88 | 36.03 | 29.96 |
| CIFAR100-C | TENT[23] @BN | 95.72 | 98.02 | 98.94 | 99.17 | 98.77 | 98.07 | 98.62 | 98.48 | 98.47 | 98.70 | 98.25 | 98.47 | 98.39 | 98.41 | 98.78 | 98.35 |
| | @*Buffer* | 90.59 | 97.43 | 97.94 | 98.21 | 98.47 | 98.39 | 98.23 | 98.27 | 98.48 | 98.30 | 98.42 | 98.50 | 97.94 | 97.96 | 98.02 | 97.66 |
| | EATA[17] @BN | 68.45 | 81.22 | 85.41 | 85.65 | 88.39 | 88.09 | 88.35 | 89.44 | 89.96 | 90.12 | 90.04 | 92.89 | 90.85 | 90.46 | 91.09 | 87.36 |
| | @*Buffer* | 60.73 | 53.85 | 56.83 | 45.00 | 60.32 | 47.54 | 46.99 | 52.27 | 52.53 | 55.84 | 45.66 | 49.71 | 57.24 | 51.57 | 59.43 | 53.03 |
| | SAR[18] @BN | 68.33 | 67.48 | 66.71 | 61.72 | 69.97 | 69.90 | 63.00 | 64.89 | 61.95 | 69.53 | 58.84 | 61.15 | 67.55 | 67.09 | 68.41 | 65.77 |
| | @*Buffer* | 58.67 | 57.73 | 59.97 | 44.46 | 59.47 | 45.96 | 45.55 | 52.25 | 51.76 | 58.34 | 44.04 | 46.72 | 53.42 | 48.46 | 55.28 | 52.14 |
| | DeYo[12] @BN | 91.82 | 97.71 | 98.03 | 97.93 | 97.83 | 98.03 | 98.03 | 98.06 | 98.37 | 98.52 | 98.58 | 98.42 | 98.38 | 98.72 | 98.82 | 97.82 |
| | @*Buffer* | 64.92 | 98.22 | 98.40 | 98.33 | 98.34 | 98.31 | 98.36 | 98.32 | 98.45 | 98.49 | 98.39 | 98.99 | 99.07 | 99.09 | 99.13 | 96.32 |
| | CMF[11] @BN | 85.15 | 95.57 | 96.32 | 96.71 | 97.55 | 97.22 | 97.51 | 97.84 | 97.80 | 97.99 | 97.81 | 97.94 | 98.16 | 98.03 | 98.52 | 96.67 |
| | @*Buffer* | 57.58 | 53.53 | 51.64 | 43.05 | 58.19 | 45.61 | 44.36 | 47.98 | 48.57 | 51.64 | 41.58 | 44.56 | 53.19 | 45.95 | 55.56 | **49.55** |
| | ROID[16] @BN | 58.32 | 56.15 | 53.83 | 45.03 | 59.96 | 45.93 | 45.69 | 51.54 | 50.88 | 55.72 | 43.87 | 46.55 | 54.59 | 48.65 | 57.56 | 51.62 |
| | @*Buffer* | 56.71 | 54.56 | 54.13 | 43.43 | 58.87 | 45.55 | 44.98 | 49.63 | 50.05 | 54.94 | 42.22 | 46.15 | 53.68 | 48.51 | 55.69 | 50.61 |

We evaluate our method under continuously evolving domains to reflect real-world deployment. This dynamic scenario challenges models to remain both responsive and stable as input distributions

gradually shift over time. Across most adaptation baselines, replacing BN updates with our *Buffer* layer yields consistently superior performance throughout the domain shift trajectory. As shown in Table.4, our method exhibits strong temporal stability and sustained accuracy, even as the input distribution drifts further from the source domain. This demonstrates that the *Buffer* layer remains effective in changing and evolving environments where traditional normalization-based approaches often struggle to keep pace.

We further note that this robustness in streaming scenarios is closely related to the *Buffer* layer's resistance to catastrophic forgetting. By preserving the original model's parameters and decoupling adaptation into an external module, the network retains critical source-domain knowledge, which in turn strengthens its adaptability over time. Our approach is thus well-suited for long-term deployment, as it preserves source knowledge and maintains adaptability over time.

### 4.2.6 Catastrophic Forgetting: *Buffer* Never Forget!

We further investigate the potential issue of catastrophic forgetting—where adaptation to the target domain degrades performance on the original source distribution—particularly in the context of CIFAR-10-W, which simulates a real-world-like continuous distribution shift (Fig.**3**). This benchmark provides a long sequence of input data with gradually evolving characteristics, making it well-suited for evaluating the stability of test-time adaptation methods. Such settings are especially relevant in practical scenarios, where models are deployed in dynamic environments and must adapt without sacrificing previously acquired knowledge. To assess this, we evaluate TENT and its *Buffer*-augmented variant on both the target data and the original source data.

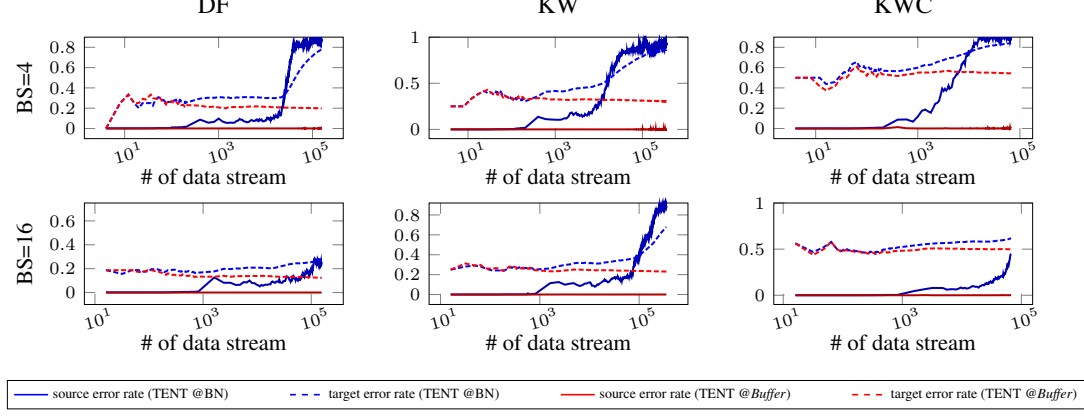

Figure 3: Catastrophic forgetting experiments on WRN28, CIFAR-10-W. Blue: TENT @*Buffer*, Red: TENT @BN.

Interestingly, we find that TENT @*Buffer* not only improves performance on the target domain but also preserves accuracy on the source domain significantly better than the original TENT. This is attributed to the fact that our method does not update the BN layers, which are essential for maintaining source-domain statistics. By isolating adaptation into the external *Buffer* layer while keeping the core network intact, our approach enables effective test-time adaptation without overwriting learned representations. Notably, even under prolonged and continuous distribution shift, our method maintains stable performance, whereas TENT suffer from performance degradation as adaptation progresses. These findings collectively demonstrate that our design is not only theoretically sound but also empirically robust in dynamic test-time scenarios.

The importance of avoiding catastrophic forgetting goes beyond preserving source-domain accuracy. In practice, we observe that once a model begins to forget the source domain, its adaptation capacity to the target domain also deteriorates over time. This suggests that forgetting disrupts the foundational representations learned during pretraining, which are crucial for effective generalization. Consequently, methods that fail to preserve source knowledge may experience compounding errors in the target domain, turning forgetting into a critical bottleneck for reliable deployment. This underscores the necessity of stable adaptation mechanisms like our *Buffer* layer that safeguard existing knowledge while enabling localized adjustments.

### 4.3 Ablation Studies

#### 4.3.1 Design Exploration of the *Buffer* layer

Our previous experiments demonstrate that attaching a *Buffer* layer to a pre-trained model, without modifying the backbone itself, enables effective test-time adaptation. While this decoupled adaptation strategy has proven successful, it remains unclear which architectural configurations of the *Buffer* layer best support this behavior. As illustrated in Fig. 4, the design space of the *Buffer* layer includes multiple options, ranging from 1×1 and 3×3 convolutions to the inclusion of BN, as well as variations in their placement within the network. In this work, we systematically explore several architectural variants (e.g., combinations of convolutions with learnable scaling factors $\alpha$ and $\beta$, optional BatchNorm) to uncover effective configurations. All ablation experiments in this section are conducted using **TENT @*Buffer***, where only the *Buffer* layer is updated during test-time adaptation while all networks including BN layers remain frozen. As shown in Table.5 and Table.6, we further examine how different structural choices and insertion locations of the *Buffer* layer impact adaptation performance. Our goal is not to propose a single optimal design but rather to investigate what aspects of localized *Buffer* module contribute most to adaptation performance. This exploration is motivated by the intuition that different layers of a pre-trained model may require distinct forms of correction, and thus, a one-size-fits-all *Buffer* structure may not be sufficient.

Empirically, we find that a dual-path configuration combining 1×1 and 3×3 convolutions (Module ④), applied after the activation layer (iii), achieves a strong balance, yielding robust performance across diverse settings. In all (iii)-type placements, Module ④ consistently outperforms other designs. However, for (ii)-type placements, the optimal configuration appears to be batch-size dependent: when the batch size is small, a single 1×1 convolution performs best, whereas for larger batch sizes, the combined 1×1 and 3×3 structure yields superior results. This suggests that the best module design depends on both its complexity and the batch size used during adaptation.

Building on the architectural choice identified as most effective in Table.5 (Module ④), we further investigate where in the network this *Buffer* layer should be placed. As shown in Table.6, we analyze the effect of inserting the *Buffer* layer at different stages of the backbone—early, middle, or late (a,b,c)—under varying batch size regimes. Interestingly, we observe that the optimal placement of the *Buffer* layer is strongly influenced by the batch size. When the batch size is small, attaching the *Buffer* layer only at the early stages yields the best results, suggesting that correcting low-level features is critical under unstable batch statistics. In contrast, with larger batch sizes, combining *Buffer* layers at both early and middle stages consistently leads to better performance.

Table 5: Block-level Error Rates.

| | Block-unit Results | | | | | | | | | | | | | | | |
|---|---|---|---|---|---|---|---|---|---|---|---|---|---|---|---|---|
| | CIFAR10 | | | | | | | | CIFAR100 | | | | | | | |
| | BS4 | | | | BS128 | | | | BS4 | | | | BS128 | | | |
| | ① | ② | ③ | ④ | ① | ② | ③ | ④ | ① | ② | ③ | ④ | ① | ② | ③ | ④ |
| (i) (Conv2D) | 29.73 | 29.57 | 31.63 | 29.36 | 20.38 | 19.53 | 20.50 | 19.44 | 48.31 | 47.92 | 51.68 | 47.85 | 35.20 | 33.23 | 34.81 | 33.18 |
| (ii) (BN2D) | 29.25 | 29.26 | 30.08 | 29.63 | 20.22 | 18.84 | 19.84 | 18.73 | 47.97 | 48.14 | 49.19 | 48.03 | 35.01 | 32.62 | 34.61 | 32.48 |
| (iii) (ReLU) | 29.36 | 29.63 | 29.95 | 29.35 | 20.28 | 19.12 | 19.49 | 19.03 | 48.29 | 48.12 | 49.47 | 47.99 | 34.45 | 32.40 | 34.21 | 32.31 |

Table 6: Module-level Error Rates.

| Configuration | | | CIFAR10 | | CIFAR100 | |
|---|---|---|---|---|---|---|
| (a) | (b) | (c) | BS4 | BS128 | BS4 | BS128 |
| ✓ | | | **29.35** | 19.03 | **47.99** | 32.31 |
| | ✓ | | 33.86 | 19.04 | 80.17 | 32.38 |
| | | ✓ | 64.64 | 20.98 | 98.43 | 75.20 |
| ✓ | ✓ | | 33.43 | **18.33** | 80.80 | **31.72** |
| ✓ | | ✓ | 63.41 | 19.97 | 98.45 | 71.16 |
| | ✓ | ✓ | 67.47 | 20.27 | 98.46 | 69.80 |
| ✓ | ✓ | ✓ | 66.94 | 19.55 | 98.43 | 67.83 |

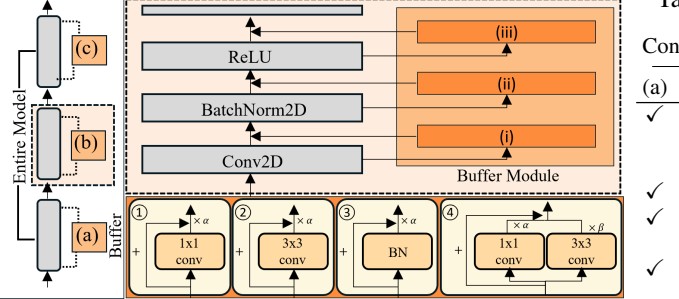

Figure 4: Module design of *Buffer* layer.

Across all configurations, we find that inserting the *Buffer* layer at the final stage of the model tends to degrade performance. This supports the hypothesis that domain shift in most cases are primarily driven by low-level distributional changes rather than high-level semantic shifts. Additionally, the batch size appears to influence not just statistical reliability but also the effective capacity of the adaptation module. Specifically, larger batches enable more stable gradient estimates, which may allow deeper or more distributed *Buffer* structures to be optimized more effectively. This insight underscores the importance of jointly considering both placement and batch dynamics when designing robust test-time adaptation modules.

### 4.3.2 Effect of $\alpha$

We analyze the role of the scaling coefficient $\alpha$ used in the *Buffer* layer, which controls the strength of test-time adaptation (Table.7). Although $\alpha$ is set as a learnable parameter, we observe that its behavior is highly sensitive to initialization. In particular, smaller initial values of $\alpha$ tend to perform better in low batch-size regimes (e.g., BS=2), preventing overfitting to noisy or unstable gradients. Conversely, larger initializations can benefit large batch sizes by enabling more aggressive adaptation. This correlation suggests that the optimal setting of $\alpha$ is not universal but instead depends on the available test-time batch size. We observe this trend consistently across both settings, with and without frozen BN layers, indicating that $\alpha$ plays a critical role regardless of whether normalization statistics are updated. These findings highlight the importance of designing a principled initialization strategy or dynamically adjusting $\alpha$ based on batch statistics for stable and effective adaptation.

Table 7: $\alpha$ sweep results on CIFAR100-C.

| | TENT @ *Buffer* | | | | | | | TENT @ BN + *Buffer* | | | | | |
| $\alpha$ | BS2 | BS4 | BS8 | BS16 | BS64 | BS256 | $\alpha$ | BS2 | BS4 | BS8 | BS16 | BS64 | BS256 |
|---|---|---|---|---|---|---|---|---|---|---|---|---|---|
| 1e-5 | 97.93 | 65.99 | 39.60 | 34.83 | 31.70 | 32.88 | 1e-5 | 98.46 | **94.51** | **78.73** | **50.46** | 32.81 | 31.24 |
| 1e-4 | **97.88** | 65.73 | 39.76 | 34.91 | 31.73 | 32.97 | 1e-4 | 98.46 | 94.56 | 78.96 | 50.70 | 32.76 | 31.24 |
| 1e-3 | 97.89 | **64.98** | 39.76 | 34.85 | 31.58 | 32.70 | 1e-3 | **98.43** | 94.71 | 78.97 | 51.22 | **32.74** | 31.18 |
| 1e-2 | 97.96 | 66.84 | **39.59** | **34.76** | **31.47** | 32.18 | 1e-2 | 98.54 | 95.07 | 80.82 | 51.81 | 32.84 | **30.90** |
| 1e-1 | 98.30 | 68.87 | 39.89 | 35.05 | 31.69 | **30.96** | 1e-1 | 98.77 | 96.95 | 89.08 | 69.32 | 36.42 | 31.11 |

## 5 Conclusion and Limitations

In this work, we proposed a lightweight and modular *Buffer* layer for source-free, fully online test-time adaptation. By decoupling the adaptation process from the backbone and targeting only the external buffer modules, our approach enables stable and architecture-agnostic adaptation even under small batch sizes or severe domain shifts. Extensive experiments across diverse datasets and TTA baselines validate the effectiveness and generalizability of the proposed method.

Despite these promising results, several limitations remain. First, the optimal architecture of the Buffer Layer has yet to be identified. Interestingly, placing the buffer after activation functions (e.g., ReLU) consistently yields better performance, but the underlying reason for this remains unclear and deserves further exploration.

Second, while we treat the scaling factor $\alpha$ as a learnable parameter, its optimization is sensitive to initialization. A principled strategy for initializing or dynamically tuning $\alpha$ is still lacking, and future work could explore meta-learned or data-aware initialization schemes.

Third, the current adaptation objective is directly borrowed from existing TTA methods, without explicitly considering the characteristics of the *Buffer* layer. Designing an optimization target tailored to the *Buffer*'s role, potentially incorporating constraints or priors reflecting its residual nature, could further enhance adaptation stability and performance.

Despite certain limitations, the proposed *Buffer* layer demonstrates notable effectiveness. It achieves strong performance without the source-dependent warm-up phase required by prior auxiliary approaches, and remains robust under small batch sizes, thereby mitigating batch dependency issues. Moreover, it effectively suppresses catastrophic forgetting, highlighting that domain adaptation can be achieved without reliance on normalization. These findings suggest a promising new paradigm for source-free adaptation.

# 6 Acknowledgement

This research was supported by Basic Science Research Program through the National Research Foundation of Korea (NRF) grant funded by the Korea government (MSIT)(RS-2025-16070382, RS-2025-02217919, RS-2025-02215070), Artificial Intelligence Graduate School Program at Yonsei University (RS2020-II201361), the KIST Institutional Program (2E33801, 2E33800), and Yonsei Signature Research Cluster Program of 2024 (2024-22-0161).

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

## A.1 Implementation Details

In this work, we compare our proposed approach against several state-of-the-art test-time adaptation (TTA) baselines. For fair comparison, we carefully reproduce each method based on official implementations and papers, and unify the training environment as much as possible. Below, we detail the optimization and hyperparameter settings used for each method.

All experiments including **TENT**, **DeYo**, **SAR**, **CMF**, and **ROID** are optimized using the Adam optimizer with a learning rate of 1e-3, $\beta = 0.9$, and zero weight decay.

**EATA** shares the same optimizer configuration as **TENT** (Adam, LR=1e-3, $\beta = 0.9$, WD=0.). Additionally, we set the Fisher regularization strength to 1.0 and the confidence margin $d$ to 0.4, following the original implementation. For the source distribution sampling, we use 2000 samples to compute the Fisher information matrix.

These unified settings ensure that performance differences primarily arise from the intrinsic mechanisms of each method, rather than discrepancies in optimization or tuning.

All experiments on **CIFAR10-C**, **CIFAR100-C**, and **ImageNet-C** [5] are conducted under **Severity 5** settings, following standard protocol to evaluate robustness under the highest level of corruption.

**CIFAR-10-W** [22] is a web-collected dataset constructed to evaluate model robustness under realistic distribution shifts. It is composed of three distinct subsets—**DF (Diffusion)**, **KW (Keyword)**, and **KWC (Keyword with Cartoon)**—each reflecting different data generation strategies and semantic characteristics. To comprehensively evaluate the generalization ability of each adaptation method, we conduct separate experiments on all three subsets.

All experiments are performed using NVIDIA RTX A6000 GPUs.

For the experiment in **Section 4.2.6 (Continuously Changing Domains)**, we use a **batch size of 16** during adaptation, which balances stability and responsiveness to gradual domain shifts.

For the analysis in **Section 4.3.2 (Effect of $\alpha$)**, we intentionally deviate from the configuration presented in Section 4.3.1. Specifically, we insert the *Buffer* layer *only after a very first single activation layer*, rather than applying it throughout the network. This simplified setting isolates the effect of the scaling parameter $\alpha$, allowing for a more controlled analysis of its influence on adaptation performance.

All *Buffer* layers are composed of **randomly initialized convolutional modules, without any pretraining**. They are optimized solely during test time, reinforcing the simplicity and modularity of the proposed approach.

Following the common practice in test-time adaptation, we configure BatchNorm(BN) layers to use target-domain batch statistics for normalization, while keeping the affine parameters frozen. This allows the model to respond to distributional shifts in the input without altering any trainable components of the normalization layers.

## A.2 Pseudo-code of Implementing *Buffer* Adaptation

---
**Algorithm 1** Test-Time Adaptation with *Buffer* layer
---
1: **Input:** test sample $\mathbf{x}$, pretrained model $\mathcal{F}_\theta$, buffer module $\mathcal{B}_\phi$, TTA algorithm $\mathcal{A}_\psi$
2: **Output:** adapted prediction $\hat{\mathbf{y}}$
3: Initialize adaptation method $\mathcal{A}_\psi$
4: Attach *Buffer* module $\mathcal{B}_\phi$ in parallel to $\mathcal{F}_\theta$
5: Freeze all parameters in $\mathcal{F}_\theta$
6: Enable gradients for $\phi$
7: $\hat{\mathbf{y}} \leftarrow \mathcal{F}_\theta(\mathbf{x}) + \mathcal{B}_\phi(\mathbf{x})$
8: $\mathcal{L} \leftarrow \mathcal{A}_\psi(\hat{\mathbf{y}})$
9: Update $\phi$ using $\nabla_\phi \mathcal{L}$
10: **return** $\hat{\mathbf{y}}$
---

The adaptation procedure with the proposed *Buffer* layer is intentionally designed to be simple and modular, as illustrated in Algorithm 1. Given any existing TTA method, the only modification required is to attach an external Buffer layer in parallel to the pretrained model and enable gradient updates for the buffer's parameters. The backbone network remains entirely frozen, ensuring that the source-domain representations are preserved, while the Buffer layer acts as a residual path that provides localized feature-level corrections during adaptation. This design not only minimizes implementation overhead but also ensures broad compatibility across different architectures and optimization schemes. As a result, our method can be easily incorporated into existing TTA pipelines with minimal code changes, without disrupting the original model structure or its pretrained functionality.

## A.3   Mixed Domains

Table 8: CIFAR100-C under mixed domain shifts.

| Dataset | | ImageNet-C |
|---|---|---|
| Models | | BS=2 |
| TENT [23] | @BN | 98.92 |
| | @*Buffer* | 98.63 |
| EATA [17] | @BN | 76.53 |
| | @*Buffer* | 76.58 |
| CMF [11] | @BN | 98.95 |
| | @*Buffer* | 83.41 |
| DeYo [12] | @BN | 96.28 |
| | @*Buffer* | 76.65 |
| SAR [18] | @BN | 76.42 |
| | @*Buffer* | 76.42 |
| ROID [16] | @BN | 95.76 |
| | @*Buffer* | 76.90 |

Considering realistic deployment scenarios, where the test distribution may shift continuously or vary across time, it is crucial to evaluate TTA methods under dynamic, mixed-domain conditions. As highlighted in [27], robust adaptation in such non-stationary environments requires models to generalize across a sequence of heterogeneous target domains without explicit domain boundaries. Inspired by this, we argue that mixed-domain evaluation provides a more rigorous and practical testbed for TTA methods, as it better reflects the challenges of real-world streaming inference. Accordingly, we include experiments under mixed corruption settings to assess the robustness and stability of our method across diverse and evolving domain shifts.

As shown in Tab.8, our method demonstrates strong performance under mixed-domain test scenarios. Across diverse corruption types and varying domain orders, the proposed *Buffer* layer consistently maintains low target-domain error while preserving source-domain accuracy, even in the absence of domain boundaries or resets. These results highlight the adaptability and stability of our approach in dynamic TTA settings, reinforcing its practical utility for real-world applications where domain shifts occur unpredictably and continuously.

## A.4   Further Experiment on Catastrophic Forgetting

As discussed in the main text, applying existing TTA approaches such as Tent to BN layers can lead to source domain forgetting, a phenomenon that becomes significantly more pronounced under relatively small batch sizes, often resulting in increased target domain error. In contrast, the proposed *Buffer* layer exhibits strong resistance to such forgetting, maintaining source performance while enabling stable adaptation to the target domain. Tab. 9 presents results on the CIFAR100-C dataset, further confirming that this robustness generalizes beyond the settings reported in the main experiments.

Table 9: Source and target errors on CIFAR100-C dataset, ResNeXT, batch size 16, Gaussian Noise.

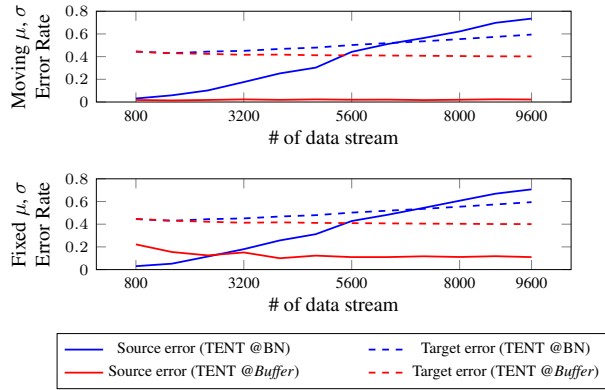

While catastrophic forgetting has been recognized as a critical challenge in TTA, prior studies have paid limited attention to establishing a standardized and fair evaluation protocol for measuring it. In particular, the question of how best to evaluate a model's retention of source-domain performance after adaptation remains largely unaddressed in the existing literature [17]. This leaves open an important methodological consideration—when assessing forgetting, should the adapted model be evaluated on source-domain data using updated source statistics (moving $\mu, \sigma$)? Or should the model

instead be evaluated using the target-domain statistics fixed during adaptation, thereby preserving the post-adaptation state (fixed $\mu, \sigma$)?

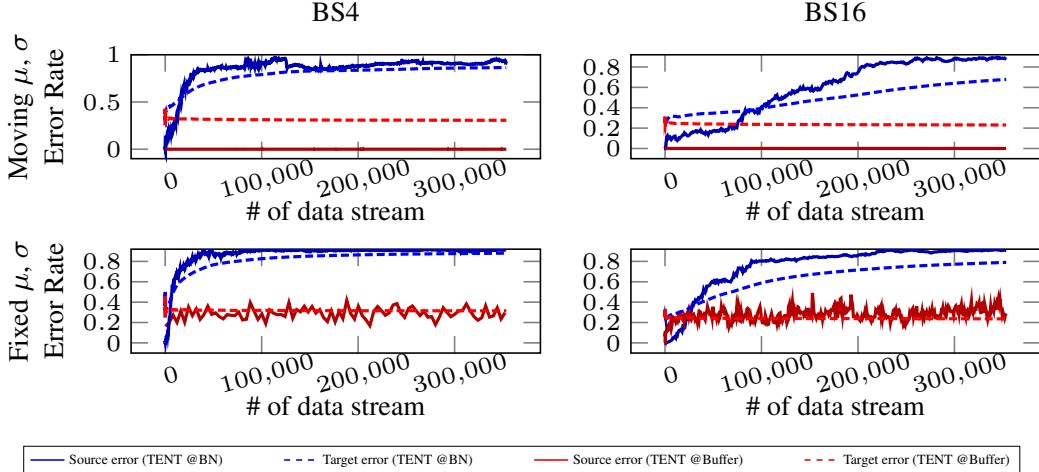

Figure 5: Catastrophic forgetting experiments on WRN28, CIFAR10-W (KW). Blue: TENT @BN, Red: TENT @Buffer.

To clarify this methodological ambiguity, we conduct experiments under both evaluation protocols. By comparing both settings, we are able to more precisely assess the impact of the *Buffer* layer on preserving source-domain performance, particularly under small batch size conditions. While both protocols yield complementary insights, we consider **the use of source-domain statistics (moving $\mu, \sigma$)** to be more practical and informative, as it reflects realistic deployment scenarios in which source-aligned calibration may still be available or preferable. Importantly, these moving statistics are collected during the standard forward pass over unlabeled source-domain data, and do not require any additional supervision or training overhead. From this perspective, using moving $\mu$ and $\sigma$ is not only cost-free but also fully consistent with test-time constraints. Accordingly, we adopt this protocol for the results presented in the main paper. As shown in Tab. 9 and Fig. 5, our method demonstrates consistently strong anti-forgetting performance across CIFAR100-C and CIFAR10-W, supporting the efficacy of the *Buffer* layer in mitigating forgetting even under challenging conditions. This effect is also clearly reflected in the source-domain results shown in Fig. 6.

This finding highlights a clear deviation from the commonly observed trade-off in TTA, wherein performance improvements on the target domain are typically accompanied by degradation on the source. The *Buffer* layer, however, achieves simultaneous gains in both domains, indicating a more favorable balance between adaptation and retention.

A central factor behind this effect is the non-intrusive and parallel architecture of the *Buffer* layer. Unlike conventional methods that adapt BN layers by updating their affine parameters, our approach avoids modifying any pretrained, learnable parameters in the backbone. In many existing methods, such updates overwrite source-domain representations, and once altered, the original alignment to the source distribution becomes difficult to recover. In contrast, the *Buffer* layer operates as a structurally independent residual branch that leverages only the input's batch statistics—mean and variance—for adaptation. Notably, this auxiliary layer can be interpreted as implicitly fulfilling the role of affine transformation in a parallel and externalized manner, enabling domain-specific modulation without interfering with the main pathway. As a result, it preserves the integrity of source-domain features while allowing effective target-domain adaptation, thereby offering enhanced robustness against catastrophic forgetting.

By maintaining a strictly parallel configuration, the *Buffer* layer allows source-domain inputs to be processed exclusively through the unmodified backbone, entirely bypassing adaptation-specific paths. This architectural decoupling provides a compelling explanation for the observed reductions in both source and target domain errors—achieved without violating the trade-off constraints that typically characterize TTA.

## A.5    Feature-level Analysis

To further validate the effectiveness of the proposed *Buffer* layer, we conducted an analysis at the feature level by examining how different adaptation strategies affect internal representations. Following the methodology of [23], we visualized the distributional statistics—specifically, the channel-wise mean and variance—of intermediate features to observe the model's response to domain shift. For this purpose, we randomly selected a subset of feature maps from the output of `stage2` (corresponding to the 18th layer in ResNexT), which captures mid-level semantics critical to downstream predictions.

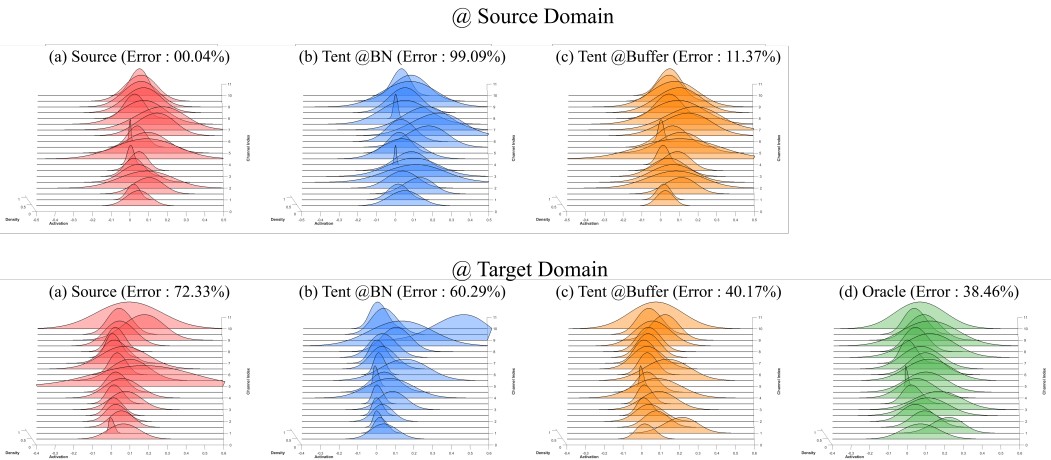

Figure 6: Feature distribution comparison across adaptation strategies on CIFAR100-C (batch size = 16). Despite being applied only at the early stage of the network, *Buffer*-based adaptation (orange) yields feature statistics (mean and variance) that are more closely aligned with the Oracle model (green), compared to TENT applied at BN layers (blue). This suggests that the Buffer layer effectively propagates adaptation signals throughout the network, enabling target-aware representations even in deep feature spaces.

We compared four adaptation settings: (a) Source (no adaptation), (b) TENT applied to BN layers (TENT@BN), (c) TENT applied to the *Buffer* layer (TENT@*Buffer*), and (d) Oracle, which is trained with full access to target-domain labels using cross-entropy loss, following the setup in [23]. All experiments were conducted on the CIFAR100-C dataset with a batch size of 16. The visualizations indicate that the feature distributions obtained from TENT@Buffer are more closely aligned with those of the Oracle model than those from TENT@BN.

This result suggests that the *Buffer* layer enables a more effective form of test-time adaptation by producing internal representations that better reflect the target-domain characteristics. The alignment with Oracle-level feature statistics provides further support for the *Buffer* layer's capacity to generalize under domain shift without requiring access to target supervision.

## A.6    Why *Buffer* layer works well?

Unlike conventional TTA methods that rely on modifying internal components of the pretrained model, our approach introduces adaptation externally—through a structurally independent *Buffer* layer. This design choice is not merely architectural; it fundamentally alters how adaptation interacts with the existing representation space. By avoiding direct updates to the backbone, the *Buffer* layer prevents destructive interference with source-domain features, a common cause of catastrophic forgetting in BN–based adaptation. Instead, the pretrained backbone remains intact, serving as a stable foundation throughout the adaptation process.

More importantly, this separation enables a distinct mode of representation learning. Instead of altering or overwriting existing features, the *Buffer* layer introduces complementary target-specific activations that coexist with the pretrained representations. This leads to an effective expansion of the class-conditional feature space, as the model learns to associate semantic concepts with a broader

range of domain-specific variations. Such behavior is conceptually aligned with findings in multi-domain and domain generalization literature, where diverse exposure to distributional shifts—without altering label semantics—has been shown to improve generalization [13]. In this light, the *Buffer* layer can be interpreted as enabling a form of adaptation-as-augmentation: it allows the model to incorporate new domain signals without sacrificing previously acquired knowledge.

In contrast to recent additive-layer approaches [21, 20] that inject adaptation modules directly into intermediate layers of the backbone, often modifying the main information flow, the *Buffer* layer remains fully external and modular. While such additive methods may retain partial structural separation, they still introduce parameter updates or architectural interference that can compromise source-domain representations. The *Buffer* layer, by comparison, performs domain-specific modulation purely through residual pathways and without altering any pretrained parameters, effectively emulating the role of affine adaptation in a parallel, non-destructive manner.

Overall, the *Buffer* layer enables a form of adaptation that sidesteps the destructive interference commonly observed in BN-based or entangled additive-layer methods. By maintaining a clean separation between the adaptation mechanism and the pretrained model, it preserves the model's original capabilities while selectively enhancing its responsiveness to target-domain signals. This balance between stability and adaptability offers a scalable and reliable foundation for test-time deployment in dynamic or continuously shifting environments.

