# OpenReview forum: "Buffer layers for Test-Time Adaptation"
_NeurIPS.cc/2025/Conference — NeurIPS 2025 poster_

### Official Review · Reviewer_S6cX · 2025-06-19

**Clarity:** 3
**Significance:** 3
**Originality:** 3
**Rating:** 4
**Confidence:** 4

**Summary:**

This paper proposes a novel concept of "Buffer Layer" for Test Time Adaptation (TTA) scenarios. This concept breaks through the traditional method of relying on normalization layer updates and addresses the problems of statistical instability of normalization layers in small batch sizes. The design of the buffer layer preserves the integrity of the pre-trained model and mitigates the risk of the catastrophic forgetting problem during online adaptation. Experimental results show good performance improvement in CIFAR and ImagNet corruption datasets.

**Questions:**

See the weaknesses. By addressing these weaknesses, the paper would provide a more comprehensive and robust evaluation of the Buffer Layer, further solidifying its contributions and potential impact in the field of test-time adaptation.

**Ethical Concerns:**

["NO or VERY MINOR ethics concerns only"]

**Final Justification:**

This work proposed a plug-and-play Buffer module, showing performance gain cross different baselines. The rebuttal adressed most of my concerns. It seems like a PEFT method for TTA in an unsupervised manner. I keep my score towards borderline accept.

**Limitations:**

yes

**Quality:**

2

**Strengths And Weaknesses:**

Strengths:
1. The proposed Buffer Layer is a plug-and-play module and can be seamlessly integrated into any pre-trained model architecture. This modularity makes it highly applicable to a wide range of models without requiring significant modifications to the original architecture.
2. The approach effectively mitigates the risk of catastrophic forgetting, which is a common issue in online adaptation scenarios. By preserving the integrity of the pre-trained backbone and isolating the adaptation process to the Buffer Layer, the model retains critical knowledge learned during pre-training. This ensures that the model does not lose its original capabilities while adapting to new domains, making it more robust and reliable in dynamic environments.
3. The Buffer Layer shows strong compatibility with existing TTA methods, consistently providing additional gains when combined with traditional normalization-based adaptation techniques. This integration enhances performance across different test-time batch sizes, demonstrating its effectiveness in both small and large batch scenarios.

Weaknesses:
1. Limited Backbone Diversity: While the paper claims that the Buffer Layer can be integrated across various architectures, the experiments are limited to ResNet-based backbones. The effectiveness of the Buffer Layer on other popular architectures, such as those based on Transformers, remains unexplored. Given the increasing prevalence of Transformer-based models in various domains, it is essential to validate the Buffer Layer's performance on these architectures to fully assess its applicability.
2. There is no detailed analysis of the computational complexity or testing time cost after integrating the proposed Buffer Layer. Understanding the additional computational overhead introduced by the Buffer Layer is crucial for practical deployment, especially in resource-constrained environments. A thorough analysis of the trade-offs between performance gains and computational costs would provide a more comprehensive evaluation of the Buffer Layer's practicality.
3. The proposed Buffer Layer shares some similarities with existing techniques such as Adapters or Side Tuning, which are primarily used in parameter-efficient fine-tuning.
4. The paper does not include experiments with a batch size of 1, which is a common scenario in real-world applications, especially in online or streaming settings.

---

> ### Author Rebuttal · Authors · 2025-07-31
>
> Thank you very much for your detailed and insightful comments. We’ve tried to answer each concern as clearly as possible and have made the corresponding revisions.
>
> ---
>
> W1 :  **Limited Backbone Diversity**
>
> R1 : We appreciate that multiple reviewers have raised the need to validate our Buffer Layer **beyond ResNet backbones**. To that end, we conducted experiments on a ViT-B/16 model: we preserved our parallel-kernel design by converting each 2D convolution into a 1D convolution over the (B, N, C) token sequence and attached the Buffer Layer immediately after each Transformer encoder block’s attention output. Despite being originally optimized for CNNs, this straightforward adaptation yields a meaningful gain on ImageNet, as shown below:
>
> The results on ImageNet-C with Batchsize 4 are as follows:
>
> | Method | Error rate(%) |
> |------------------|------------------|
> | Tent                 | 54.40 |
> | Tent @ Buffer | **52.07**|
> | CMF| 67.77 |
> | CMF @ Buffer | 66.74 |
> | SAR                | 53.59 |
> | SAR @ Buffer | 54.09 |
> | Deyo                 | 65.05 |
> | Deyo @ Buffer | 65.62 |
> | ROID                | 68.87 |
> | ROID @ Buffer | 66.02 |
>
> As the results show, the Buffer Layer also works on Transformer backbones. That said, here we have merely taken the CNN-optimized Buffer and converted its parallel convolutions into simple 1D counterparts—this is a straight-forward, proof-of-concept adaptation. A buffer module designed specifically for Transformers (e.g., one that better leverages the self-attention structure) or a more architecture-agnostic/generalized design would likely yield even stronger results.
>
> ---
>
> W2 : **Analysis of the computational complexity or testing time cost**
>
> R2 : Thank you for this excellent question. To give a sense of the overhead, we include a summary comparison in the table below.
>
>
> | Method       | # of Trainable Parameters              |                      | Vram Usage (GiB) |             | Inference Time (s) (per batch) |             |
> |--------------|----------------------------------------|----------------------|------------|-------------|-------------------------------|-------------|
> |              | WRN28 (CIFAR10)                        | Res50 (ImageNet)     | WRN28      | Res50       | WRN28                         | Res50       |
> | Tent @BN         | 18K / 36.5M (0.05%)                    | 53K / 25.6M (0.21%)  | 2.2       | 3.4       | 9.35                          | 11.15       |
> | Tent @Buffer     | 1.79M / 38.3M (4.67%)                  | 77.9K / 25.6M (0.30%)| 2.9       | 3.9        | 20.45                         | 15.28       |
> | Tent @Buffer+BN  | 1.81M / 38.3M (4.74%)                  | 131K / 25.6M (0.51%) | 2.9       | 3.8        | 20.4                          | 15.30        |
> | SAR @BN        | 18K / 36.5M (0.05%)                    | 30.6K / 25.6M (0.12%)| 2.2       | 3.3        | 19.06                         | 19.91       |
> | SAR @Buffer | 1.79M / 38.3M (4.67%)                  | 77.9K / 25.6M (0.30%)| 2.7       | 3.7        | 31.04                         | 23.49       |
> | SAR @Buffer+BN  | 1.81M / 38.3M (4.74%)                  | 108K / 25.6M (0.42%) | 2.7       | 3.6        | 32.21                         | 24.27       |
>
>
>
> As shown, integrating the Buffer Layer incurs a modest increase in number of parameters, vRAM, and latency. There is some added test-time cost—especially during adaptation—because the Buffer requires additional computation—but because only a small number of parameters is trained and updates are localized, the extra computational burden remains light. Considering the substantial accuracy gains under domain shift, we believe this trade-off is favorable in many practical settings.
>
> Making the Buffer Layer even more efficient while preserving its effectiveness (e.g., via low-rank approximations of Buffer Layers, conditional activation, or lightweight architectural variants) is a promising direction for future work.
>
> ---
>
> W3 : **Similarity to PEFT methods**
>
> R3 : **Comparison to parameter-efficient fine-tuning (PEFT) methods**
> We agree that the Buffer Layer might be seen conceptually similar to PEFT techniques such as Adapters, Side Tuning, and LoRA—each freezes (or largely preserves) a core model and adds a lightweight auxiliary component. However, the **goal and setting** differ: PEFT methods are typically applied with **labeled data** to fine-tune for a known downstream task, whereas our Buffer Layer is designed for **unsupervised test-time adaptation**, correcting domain shift on-the-fly using only unlabeled target samples. Architecturally, while LoRA and Adapters introduce low-rank or bottleneck transformations inside or alongside weight matrices, and Side Tuning adds a parallel branch with supervision, our Buffer Layer uses parallel convolutional residual corrections (1×1 + 3×3) that are conditioned and scaled to gently adjust representations without overwriting them.
>
> To make this distinction concrete, we will include two sets of controlled empirical comparisons: one with **LoRA applied to convolutional backbones** (original version and matching the depth and insertion points of our buffer and using the same TENT entropy-minimization objective for test-time adaptation) and another with **LoRA on transformer-based backbones** (e.g., ViT), again under the same unsupervised adaptation protocol.
>
> | Method      | Config     | BS2 |BS4 | BS16 |
> |-------------|------------|------------|-------------|--------------|
> | Tent| LoRA(early)    | 39.20       | 29.85       | 21.02        |
> | |  LoRA(full)       | 87.13      | 66.42       | 24.39        |
> |         | @BN         | 82.56      | 51.36       | 23.12        |
> |             | @Buffer     | **37.05**  | **29.11**       | **20.32**    |
>
> The table above reports ConvLoRA results on WRN28 / CIFAR-10 under the same test-time entropy-minimization objective used for our Buffer Layer. As shown, Buffer outperforms ConvLoRA, despite the superficial structural similarity of both methods (freezing the backbone and adding a small adaptation module). We believe the gap arises because the Buffer Layer’s parallel convolutional residual branches can apply localized, input-conditioned corrections—which is crucial for handling spatially varying domain shifts—whereas ConvLoRA’s low-rank weight updates are more global and less capable of capturing those fine-grained distortions. In short, sharing the high-level idea isn’t enough: the specific design of the Buffer Layer appears better suited to the test-time domain adaptation setting, as evidenced by the performance difference.
>
>
>
> | Method | Error rate(%) |
> |------------------|------------------|
> | Tent                 | 54.40 |
> | Tent @ Buffer | **52.07**|
> | CMF| 67.77 |
> | CMF @ Buffer | 66.74 |
> | SAR                | 53.59 |
> | SAR @ Buffer | 54.09 |
> | Deyo                 | 65.05 |
> | Deyo @ Buffer | 65.62 |
> | ROID                | 68.87 |
> | ROID @ Buffer | 66.02 |
>
> The table above shows results on ImageNet-C with ViT-B/16 (batch size 4); it again performs well and exhibits the same behavior described in the main paper. This illustrates that, although the Buffer Layer may superficially resemble PEFT, applying PEFT directly to TTA does not yield comparable results.
>
> ---
>
> W4 : **Evaluation on Batchsize 1**
>
> R4 : Thank you for this important question. The reason we did not include vanilla BS=1 results for most of our reported baselines (TENT, EATA, DeYO, etc.) is that they are BatchNorm-based, and BatchNorm collapses in the extreme BS=1 regime—baseline accuracy on corrupted CIFAR-10-C and CIFAR-100-C falls to near chance (e.g., ~10% and ~1%, respectively)—so there is effectively no meaningful signal for the Buffer Layer to build on in those settings. In other words, when the underlying adaptation mechanism fails catastrophically due to BN’s instability at batch size 1, simply stacking a buffer on top does not salvage it, which is why those pure BS=1 results were not included.
>
> To validate, we do have two complementary scenarios where BS=1 is feasible and the Buffer Layer still delivers gains:
>
> **ResNet50 with GroupNorm** : GroupNorm is relatively insensitive to batch size because it normalizes across channels/groups rather than across the batch. In this configuration, TENT (and by extension our adaptation pipeline) remains functional at BS=1, and we observe that attaching the Buffer Layer further improves performance, confirming that the method can work in true single-sample online settings when the normalization backbone is stable.
>
> |  Resnet50 (GN) (Error rate %) ↓ |  |
> |-----|-----|
> | | BS1|
> |Tent @GN | 85.30|
> |Tent @Buffer | **71.58**|
>
>
> **Semantic segmentation** : As also highlighted in the TENT paper, segmentation naturally provides dense, per-pixel supervision-like signals even with batch size 1, since each image yields many spatial locations. We evaluated a Deeplabv2 model under a day-to-night shift on the CARLA dataset, attaching the Buffer Layer to the encoder. Despite BS=1, we observed a meaningful increase in mIoU, demonstrating that the Buffer Layer can be effective in real-world streaming/online scenarios where each “batch” is a single image with rich internal structure.
>
>
> | | Deeplabv2 (mIOU ↑) |
> |-----|-----|
> | | BS1|
> |Source (No Adaptation) | 52.75|
> |Tent @BN | 55.98|
> |Tent @Buffer | **56.68**|
>
>
> Together, these results indicate that although standard BN-based classifiers collapse at BS=1, the Buffer Layer can still be effective in single-sample settings when the normalization or task structure supports stable updates—making batch size 1 a valuable direction for further study. Thank you for this suggestion. The insightful questions provided have truly enriched the paper and helped us think more deeply about the practical aspects of TTA.

---

> > ### Author Response · Authors · 2025-08-01
> > **Correction of an Error**
> >
> > Dear Reviewer S6cX,
> >
> > Thank you again for taking the time to carefully review our paper. We would like to correct an error in our response to your question W3 (“Similarity to PEFT methods”). Our intention was to present two results ( 1.ConvLoRA and 2.Transformer-LoRA) so that we could demonstrate not only the conceptual relationship to PEFT but also the empirical performance differences when PEFT is applied in the TTA setting. However, due to our mistake, the table that followed ConvLoRA was not the Transformer-LoRA result but the table used for your question W1 (“Limited Backbone Diversity”). We sincerely apologize for this oversight and for any confusion it caused.
> >
> > Below is the table we originally intended to include:
> >
> > | Method      |BS4 | BS16 | BS64 |
> > |-------------|-----------|------------|------------|
> > | Source(No Adaptation)     | 59.51     | 59.51      | 59.51      |
> > | LoRA + Tent | 95.27     | 86.67      | 71.44      |
> > | Tent @ Buffer       | 54.48     | 56.88      | 58.88      |
> >
> > As shown, when applying the LoRA structure on ViT-B/16 while keeping only Tent’s entropy-based objective for TTA, LoRA in fact fails to provide the expected adaptation benefit.
> >
> > We apologize again for the misunderstanding and the resulting confusion. If you have any further questions or would like additional clarification, please feel free to ask. Thank you once more for your valuable feedback.

---

> > > ### Comment · Reviewer_S6cX · 2025-08-04
> > >
> > > Thank you for the response and clarification. The rebuttal adressed most of my concerns. I suggest to include more related work and discussion for PEFT and latest TTA methods which also insert lightweight trainable component.

---

> > > > ### Author Response · Authors · 2025-08-07
> > > >
> > > > Thank you for your thoughtful response and suggestions.
> > > >
> > > > We appreciate your recommendation to include more related work and discussion on PEFT (Parameter-Efficient Fine-Tuning) and recent TTA methods that introduce lightweight trainable components. We agree that this is an important and rapidly evolving area, and we will make sure to expand the related work section and include a more detailed discussion on how our Buffer Layer relates to and differs from these approaches in the final version of the paper.
> > > >
> > > > Thanks again for your valuable input.

---

### Official Review · Reviewer_kqmB · 2025-06-20

**Clarity:** 4
**Significance:** 4
**Originality:** 3
**Rating:** 5
**Confidence:** 4

**Summary:**

Conventionally, test-time adaptation methods often update the parts of the model parameters, such as the affine parameters in the normalization layers, or update the entire parameters, but these methods suffer from the problem of ineffectiveness when the batch size is small and the catastrophic forgetting problem. This paper addressed these problems by introducing a light-weight network module, called Buffer Layer, and updating it while freezing the original network parameters.

**Questions:**

The paper was clear and easy to understand, so I don't have any additional questions. I have summarized my concerns in the "weakness".

**Ethical Concerns:**

["NO or VERY MINOR ethics concerns only"]

**Final Justification:**

The author's rebuttal has largely addressed my main concerns, such as applicability to ViT, comparison to other PEFT methods, etc., so I would like to maintain my initial “Accept” rating with higher degree of confidence.

**Limitations:**

yes

**Quality:**

3

**Strengths And Weaknesses:**

**Strengthes**
1. The proposed method is simple and effective, and　it has a significant contribution to this research area: This paper proposed a very simple method that adds lightweight modules to the base network, yet it effectively addresses the problems of the conventional approaches. In addition, in the experiments, the proposed method demonstrated its effectiveness when introduced to several existing methods, which indicates the broad compatibility of the method. These results demonstrate the potential to completely replace the conventional approach of training only normalization layers, and I believe that this will be a significant contribution to this field.
2. Comprehensiveness of the Experiment: The experiments and analysis were comprehensive and adequately supported the claims of this study.
3. Paper organization: This paper is well-written and easy to follow.


**Weaknesses**
1. Comparison to parameter-efficient fine-tuning (PEFT) methods: The proposed method is similar to the existing PEFT methods (e.g., LoRA), which update only a small number of additional parameters. It is necessary to discuss the difference between these methods and the proposed method.
2. Applicability to the transformer-based networks: The proposed method seems to be designed for the convolution-based networks like ResNet, but there remains a concern about the applicability to the non-convolution-based networks like Vision Transformer. The modern neural network is mainly built upon the transformer-based architecture, and thus, it is necessary to investigate the applicability to such transformer-based models. In addition, this is related to the weakness 1., LoRA is applicable to the transformer-based models, so it is interesting to compare the effectiveness of the proposed method and LoRA.
3. Analysis about the forgetting: In the experiments, it is shown whether the proposed method is effective against forgetting, but there seems to be a lack of discussion or analysis of why it is effective. The authors explained this is because the original model parameters are frozen, but I think the activations inside the network are changed by the buffer layer training, and this change may cause catastrophic forgetting. I believe that analyzing the mechanism by which the proposed method prevents forgetting will lead to a further understanding of the proposed method.
4. Minor points:
    + Typo in l135: Sec Sec. -> Sec.
    + Figure 3 is not referred to in the main text.
    + In Figure 3, I'm not sure what "DF", "KW", and "KWC" mean.

---

> ### Author Rebuttal · Authors · 2025-07-31
>
> We’re grateful for your detailed feedback. It helped us improve the manuscript, and our responses to your points are below
>
> ---
>
> W1 :  **Comparison to parameter‑efficient fine‑tuning (PEFT) methods**
>
> R1 : While both our Buffer Layer and PEFT methods like LoRA freeze pretrained weights and learn only a few extra parameters, they contrast sharply in purpose and setting: PEFT leverages labeled downstream data for supervised fine‑tuning on a known task, whereas our Buffer Layer performs unsupervised test‑time adaptation, updating on unlabeled samples to correct domain shifts at inference. Architecturally, LoRA injects low‑rank updates into existing weight matrices, while we append lightweight parallel 1×1 and 3×3 convolutional branches alongside the backbone. We have provided **a detailed LoRA comparison in R2** and will expand the Introduction and Related Work to underscore these key differences.
>
> ---
>
> W2 : **Applicability to the transformer-based networks**
>
> R2 : Thank you for highlighting the importance of evaluating on Vision Transformers and comparing to PEFT methods in that setting. First, to demonstrate that our framework can be effectively applied to ViT, we converted each 2D convolutional branch into a simple 1D convolution over the (B, N, C) token sequence of a ViT‑B/16 backbone. The results on ImageNet-C with Batchsize 4 are as follows:
>
> | Method | Error rate(%) |
> |------------------|------------------|
> | Tent                 | 54.40 |
> | Tent @ Buffer | **52.07**|
> | CMF| 67.77 |
> | CMF @ Buffer | 66.74 |
> | SAR                | 53.59 |
> | SAR @ Buffer | 54.09 |
> | Deyo                 | 65.05 |
> | Deyo @ Buffer | 65.62 |
> | ROID                | 68.87 |
> | ROID @ Buffer | 66.02 |
>
> As the table demonstrates, the Buffer Layer shows its effectiveness on ViT-B/16, confirming that the approach generalizes beyond convolutional backbones.
>
> For the LoRA‑based TTA variant, we initialized all LoRA adapters to zero and leveraged TENT’s entropy‑based update mechanism—mirroring our buffer update strategy but within the LoRA parameterization. The results are as below :
>
> | Method      |BS4 | BS16 | BS64 |
> |-------------|-----------|------------|------------|
> | Source (No Adaptation)     | 59.51     | 59.51      | 59.51      |
> | LoRA + Tent | 95.27     | 86.67      | 71.44      |
> | Tent @Buffer     | 54.48     | 56.88      | 58.88      |
>
>
> As shown, TENT\@Buffer outperforms the LoRA-based adaptation on the **same Transformer backbone**, confirming our buffer strategy’s competitiveness **beyond CNNs**. While structurally and conceptually similar to LoRA in that both freeze the backbone and add a small adaptation module, the test-time adaptation setting exposes a clear difference: our parallel convolutional residual branches with controlled corrections are more efficient and better suited to correcting domain shift than LoRA’s low-rank weight updates, making it a distinct (and in this case more effective) design. We agree that a Transformer-specific buffer module could yield further gains. Thank you again for this insightful suggestion.
>
> To further discuss and verify whether the positional preference for Buffer Layer insertion that we discussed in the main manuscript carries over to Transformers, we conducted an additional ablation study on ViT-B/16, varying where the Buffer is attached (e.g., at different depths).
>
> | BS4 | Error rates|
> |------------------|------------------|
> | Layer1 | 52.23|
> | **Layer2** | **52.07**|
> | Layer3 | 54.73|
> | Layer4 | 55.23|
> | Layer5 | 55.23|
> | Layer6 | 54.28|
> | Layer7 | 55.68|
> | Layer8 | 55.50|
>
> As in Table 6 of the main text, performance varies with insertion location: attaching the Buffer at early depths produces the largest gains, while later placements yield smaller improvements. This finding both increases consistency with the patterns reported in the main manuscript and is an interesting result in its own right—it suggests that the insertion dynamics we observed in CNNs have a similarity in Transformer architectures, reinforcing the generality of the design.
>
> ---
>
> W3 : **Analysis about the forgetting**
>
> R3 : The reviewer’s point is well taken, and we’ve been exploring the same question: if the buffer injects corrections in parallel, why doesn’t that induce forgetting? We believe the answer is that the Buffer Layer behaves more like a **conditional residual adapter** than an overwrite. Its output is added to the backbone’s output in a data-dependent way, with a learned scale, so the original representation is preserved and only gently adjusted based on the input: on **target** inputs the residuals actively compensate for domain shift, whereas on **source** inputs the added corrections tend to be near-neutral and do not significantly perturb the representation. Because the backbone is frozen, upstream features remain fixed, and the buffer’s influence is localized and tempered, avoiding wholesale drift.
>
> Supporting this, Supplementary Figure 2 shows that after adaptation the model’s target feature distribution moves closer to the oracle (as expected), while the source data distribution largely retains its original shape. Even though the buffer adds a residual correction, it does not collapse or significantly distort the source representation, which is consistent with a conditional, controlled adjustment rather than a destructive change. This behavior, improved target alignment with minimal impact on source, suggests why catastrophic forgetting is mitigated in our approach.
>
>
> ---
>
>
>
> **[Minor Points]**
> - Actually, it is "**See** Sec", Not "Sec Sec". We will change this sentence to such as 'Note Sec~' etc.
> - Thank you so much for your detailed comments. We will add referring  Fig.3. in the main text.
> - Sorry for the inconvenience. The CIFAR‑10‑W dataset is a realistic‑domain‑distribution benchmark for evaluating domain shift, comprising KW: keyword search domains, KWC: keyword with color domains, and DF: diffusion‑generated domains; we will add detailed descriptions in the main text.
>
> We’re grateful for your careful and insightful review; it’s been extremely helpful in strengthening the paper. Thank you.

---

> > ### Comment · Reviewer_kqmB · 2025-08-02
> >
> > Thank you for the author responces to my comments. These responces have satisfactorily resolved all my concerns, so I would like to keep my tendency to acceptance.

---

> > > ### Author Response · Authors · 2025-08-04
> > > **Thank you!**
> > >
> > > Thank you for your careful review and positive feedback. We're glad our responses addressed your concerns!

---

### Official Review · Reviewer_RLdc · 2025-06-30

**Clarity:** 4
**Significance:** 3
**Originality:** 3
**Rating:** 4
**Confidence:** 4

**Summary:**

The paper investigates the problem of test-time adaptation (TTA) from the angle of which parameters to update, instead of how to update them. The proposed method builds on the idea that most approaches update the batch norm (BN) parameters of the model, however these parameters are highly sensitive to small batchsizes. Similar to EcoTTA, the method keeps the model untouched and instead adds a Buffer module on top of initial layers of the network, which is adapted instead of the BN parameters using the usual entropy or augmentation consistency losses. In a comprehensive set of experiments, authors show that adding their Buffer Module on top of existing TTA boosts their accuracy under different types of data corruption settings.

**Questions:**

Please see the weaknesses above. Technically, my current evaluation is closer to a Weak Accept. I would be willing to upgrade my score to Accept if my concerns are addressed by authors.

**Ethical Concerns:**

["NO or VERY MINOR ethics concerns only"]

**Final Justification:**

Although I initially considered upgrading my score to 5, I was not satisfied with the authors' answer regarding the comparison to other approaches. In particular, they did not compare their method with the Layer norm variant of SAR, as suggested in my comment, and instead reported results for the weaker Batch norm and Group norm variants.

**Limitations:**

Yes

**Quality:**

3

**Strengths And Weaknesses:**

Strengths:

* The method is simple but effective. The Buffer module can be added on top of most TTA approach, without any modification, to improve their robustness to corruptions at test time.

* As the method is rather simple to explain, the paper instead builds on evaluating it under a broad range of settings. Overllal, results of these experiments are convincing.

* The paper is quite well written. Thanks !

Weaknesses:

* Although mentioned very quickly in Section 2.1, the paper does not properly present the efforts of previous works to address the instability of BN update for small batchsizes. Those include SAR which replaces the BN by Layer Norm or Group Norm, and RoTTA which updates the statistics using an exponential moving average.

* Some of the reported results do not match with those published in previous works. For example, in Table 3, authors report a classification error of 94.34% for SAR in the BN=2 and GN setting, while the error reported in the SAR paper for the same setting is 65.5%.

* Moreover, some comparisons in the experiments are a bit unfair. For instance, the Layer Norm variant of SAR  (on a ViT backbone) achieves much stronger results, but authors only evaluate their method on CNN backbone. Furthermore, EcoTTA is not included in the comparison since it requires initializing the external module with source data. However, a better picture would be obtained by adding this method, with this initialization and also using a random initialization like in the paper.

* As the module starts with randomly initialized weights, its prediction for the first test samples might be very noise. This problem is alleviated by employing a small alpha to reduce the module's initial influence. I believe this problem should be discussed with greater depth in the paper.

---

> ### Author Rebuttal · Authors · 2025-07-31
>
> We’re grateful for your thoughtful feedback, which has helped us refine our manuscript. Below, we provide our detailed responses to your comments.
>
> ---
>
> W1 :  **Lack of presentation of previous works on instability of BN updates with small batch sizes**
>
> R1 : We thank the reviewer for highlighting the important work on stabilizing BN in low‑batch regimes, notably SAR, which replaces BatchNorm with LayerNorm or GroupNorm, and RoTTA, which smooths BN statistics via an exponential moving average. We fully recognize the value of these approaches. In our revision, we will expand Section 2.1 with a detailed survey of SAR’s normalization swap and RoTTA’s EMA‑based statistic update, and clearly contrast them with our Buffer Layer strategy—where we preserve BN and instead introduce lightweight parallel modules for adaptation. This addition will both credit these contributions appropriately and strengthen our related‑work discussion.
>
> ---
>
> W2 : **Discrepancy in SAR results**
>
> R2 : We apologize for the confusion. The 65.5% error reported in the SAR paper refers to a **ResNet50 with GroupNorm**, whereas our Table 3 uses the **ResNetV2** backbone from Big Transfer [Kolesnikov et al., ECCV 2020], which—despite **also employing GN**—is a **different architecture**. Our purpose of including ResNetV2 results was to demonstrate the broad applicability of our method across diverse backbones.
> To resolve this misunderstanding and clarify our comparisons, we have reproduced the SAR setting on ResNet50 (GN), with the following results:
>
> | Method | BS 1 | BS 2 | BS 4 | BS 16 |
> |------------------|------------------|-------------------|-------------------|-------------------|
> | Tent  @ GN               | 85.30 | 80.49 | 77.37 | 71.60 |
> | Tent @ Buffer | 71.58 | 78.48 | 71.58 | 69.66 |
> | SAR @ GN | 66.03 | 67.78 | 66.75 | 67.89 |
>
>
> Although the SAR paper only reports results at batch size 1, our reproduction (**66.03 vs. 65.5 on SAR paper Table 4**) is in close agreement, supporting the validity of the reproducibility and the reliability of our evaluation pipeline. We also applied our Buffer Layer on this ResNet50(GN) setup and observed a consistent error reduction on Tent, underscoring the method’s effectiveness across architectures. We appreciate this insightful question and will update to clearly distinguish between the different backbones and their respective results.
>
> ---
>
>
> W3 : This question touches on two separate issues, so to ensure clarity we will address them in two parts:
>
> W3-1 :  **SAR on CNN(BN)/ViT(LN) : Fair Evaluation issue**
>
> R3-1 : We appreciate the reviewer’s observation that SAR was designed specifically for GN or LN (on Vision Transformers) and that evaluating exclusively on CNNs with BN may seem unfair. We have already included **ResNetV2(GN)** comparisons in Table 3 of the main manuscript. To further demonstrate our method’s generality beyond CNNs, we adapted the Buffer Layer to a **ViT‑B/16** backbone, by converting the Buffer Layer with its 2D convolutional path into a simple 1D convolution over the (B, N, C) token sequence. We emphasize that, while our original 1×1 + 3×3 design targets 2D feature maps and **may not be optimal** for Transformers, this proof‑of‑concept shows extensibility. The results below confirm that TENT@Buffer (1D conv on ViT‑B/16) outperforms the SAR LN variant at Batchsize 4. (To further validate reproducibility, we began by verifying that our reproduction of the SAR setting at batch size 1 closely matches the 65.5% reported in the SAR paper).
>
> | Method | | Error rate(%) |
> |------------------|------------------|------------------|
> | Tent |@BN                | 54.40 |
> | |@ Buffer | **52.07**|
> | CMF| @BN| 67.77 |
> | |@ Buffer | 66.74 |
> | SAR|@BN    | 53.59 |
> | | @ Buffer | 54.09 |
> | Deyo|@BN                 | 65.05 |
> | |@ Buffer | 65.62 |
> | ROID      |   @BN       | 68.87 |
> | |@ Buffer | 66.02 |
>
>
> We think that a transformer‑tailored buffer module might yield even greater gains, and this remains an exciting direction for future research. We will update Section 3.2 to include this comparison and clearly highlight our method’s broader applicability. Thank you for this excellent question, which has allowed us to rigorously validate our approach once again.
>
> W3-2 :  **EcoTTA with/without Initialization**
>
> R3-2 : Thank you for this valuable suggestion. As noted, EcoTTA requires source‑data warming, which is why we did not include it originally. However, we agree that presenting both the warm‑up and fully source‑free variants of EcoTTA will enrich our experimental comparison. Below we report EcoTTA error rates on CIFAR‑10‑C (WRN28 backbone) both with source‑data warm‑up and without (fully source‑free), alongside our TENT@Buffer results:
>
> |                      | **BS2** | **BS4** | **BS16** |
> |----------------------|--------:|--------:|---------:|
> | EcoTTA, random init  | 88.64 | 88.15 | 83.26 |
> | EcoTTA, warmup       | 82.72   | 53.78   | 21.95    |
> | TENT @buffer         | 37.05   | 29.11   | 20.32    |
>
> As shown, TENT@Buffer outperforms EcoTTA across all batch sizes, even when EcoTTA leverages source‑data warm‑up. This mirrors the EcoTTA authors’ observation that small‑batch regimes degrade their stability—necessitating an additional ‘adaptBN’ module—whereas our Buffer Layer achieves robust, stable performance on small batches without any source data, demonstrating a clear advantage.
>
> ---
>
> W4 : **Initial noise from random buffer weights**
>
> R4 : Thank you for this excellent insight and for giving us the opportunity to delve deeper into our buffer design. In addition to the small‑batch gradient noise you highlighted, we believe there are two primary sources of noise in our adaptation procedure: (1) the inherent variance from very small batch updates, and (2) the random initial weights of the buffer itself.
> To isolate the latter, we compared our standard random initialization against a zero‑initialized buffer across a range of alpha values (1e‑5, 1e-3, 1e‑1). As summarized in the table below, the zero‑init buffer shows minimal sensitivity to α, maintaining stable error rates, whereas the randomly initialized buffer’s error varies by nearly 1 pp across the same α sweep. Surprisingly, at large batch regime, the zero‑initialized buffer even outperforms our original random‑init results—underscoring that initialization‑induced noise can be as impactful as batch‑size effects. These observations echo similar findings in LoRA and other PEFT works and point to initialization strategy as a rich direction for future exploration.
>
> | init_method | lr        | BS2     | BS16     | BS256    |
> |-------------|-----------|---------|----------|----------|
> | random| 1.00E-05  | 74.85   | 35.87    | 33.11    |
> |             | 1.00E-04  | **74.67** | 35.85    | 33.10    |
> |             | 1.00E-03  | 74.98   | 35.87    | 33.10    |
> |             | 1.00E-02  | 75.00   | 35.81    | 33.09    |
> |             | 1.00E-01  | 74.87   | **35.80** | **32.91** |
> | zero| 1.00E-05  | 75.07   | 34.34    | 31.77    |
> |             | 1.00E-04  | 75.12   | 34.32    | 31.79    |
> |             | 1.00E-03  | 75.24   | 34.27    | 31.82    |
> |             | 1.00E-02  | 75.19   | 34.39    | 31.80    |
> |             | 1.00E-01  | **74.76** | **34.21** | **31.45** |
>
> Thank you again for this valuable feedback, which helped make the paper substantially richer. If you have any further comments or follow-up questions, we would be grateful to hear them.

---

> > ### Comment · Reviewer_RLdc · 2025-08-04
> > **Thanks for the clarifications**
> >
> > I thank the authors their detailed response to my comments. However, I have some remaining comments and questions regarding the comparison to SAR:
> >
> > * For Table 3 in the paper, I think it would be better to have a single backbone (ResNet50) for comparing BN against GN. The table could also include the results for BS=1 which is supported in methods like SAR. Finally, the word 'BN' in the second column could be replaced by 'BN/GN' since the table reports results for both settings.
> >
> > * For the results on a ViT backbone, why can't you reconstruct a 2D feature map from the tokens and then apply a standard convolution as in your paper?  Also, why not compare against SAR (LN) in the new Table, since that is the most common type of normalization for this architecture? Note that in Table 2 of the SAR paper, this setting achieves an error rate of 43.3% for ImageNet-C (severity 5, BS=1), which is considerably lower than the 53.3% reported above.

---

> > > ### Author Response · Authors · 2025-08-07
> > > **Reply (1/2) of the comments**
> > >
> > > First of all, thank you for your additional comments.
> > >
> > > We have been sincerely conducting additional experiments to respond faithfully to your suggestions. Apologies if our reply comes with a slight delay. We will address your questions one by one below.
> > >
> > > **Q1: For Table 3 in the paper, I think it would be better to have a single backbone (ResNet50) for comparing BN against GN. The table could also include the results for BS=1 which is supported in methods like SAR. Finally, the word 'BN' in the second column could be replaced by 'BN/GN' since the table reports results for both setting.**
> > >
> > > R1: Thank you very much for this insightful comment. As you suggested, rather than using ResNetV2, it makes more sense to **fix the backbone to a single architecture** and compare BN and GN within **ResNet50**. We also agree that the second column should be labeled as 'BN/GN' to reflect both settings. We are currently finalizing the revised table.
> > >
> > > However, while addressing the second part of your suggestion regarding the performance gap with the SAR paper, it required additional setup — we decided to switch the entire codebase and reconfigure all experimental settings, which took considerable time. The details of this process are explained in our response to **R2 below**. Therefore, the detailed metrics can be different from the proposed one on our previous rebuttal. However, we have double-checked the fair evaluation.
> > >
> > > For now, we have included part of the Table 3. We will fill in the remaining entries and share the fully completed version before the end of the discussion period.
> > > Below is the updated right-hand side of Table 3 **(ResNet50-GN) (in terms of Error rates (%))**:
> > > We have also included the results of the Batch size 1, so as to address your concern. The Table.3. on the manuscript will be replaced to one with the below.
> > >
> > > | Method     |BS1 |BS4 | BS16 | BS64 |
> > > |-------------|-----------|------------|------------|------------|
> > > | Source    | 69.10 | 69.10     | 69.10      |  69.10      |
> > > | Tent @ GN | 78.32    | 78.31    | 78.27    |  78.23    |
> > > | Tent @ Buffer | **72.20**    | **72.13**    | **72.01**    |  **70.72**    |
> > > | Tent @ GN+Buffer | 80.10    | 79.58    | 78.58    |  **77.61**    |
> > > | | | | |
> > > | SAR @ GN | 65.57    | 65.61    | 63.55  |  64.31    |
> > > | SAR @ Buffer | 68.53    | 68.77    | 68.34    |  67.60    |
> > > | SAR @ GN+Buffer | **64.72**   | **65.13**    | **59.31**    |  **63.77**    |
> > >
> > > We have checked that Res50-GN results above align with those reported in the SAR paper.

---

> > > ### Author Response · Authors · 2025-08-07
> > > **Reply (2/2) of the comments**
> > >
> > > **Q2: For the results on a ViT backbone, ...**
> > >
> > > R2  : Let us respond to your points one by one.
> > >
> > > - **2D Feature Map ViT Buffer**:
> > > We appreciate your excellent suggestion. Instead of reconstructing the Buffer architecture in a 1D manner, we extracted the intermediate tokens from the Transformer with shape (B,N,C) ,removed class token (B, N-1, C), and reshaped into 2D-feature map as (B,C,H,W). This allowed us to apply our **Buffer layer identically** to how it was originally used in CNN-based architectures. Interestingly, this adjustment led to even better results, as presented in the updated **table below**. This experiment confirmed that our Buffer module can be universally applied even to Transformer-based architectures.
> > >
> > > - **Discrepancy with the SAR Paper** :
> > > We deeply investigated the discrepancies between our results and those reported in the SAR paper, and were **finally able to resolve the issue**. Below are the key differences we identified between our setup and the SAR implementation:
> > >
> > > **Reasons for the Discrepancy**:
> > >
> > > 1. **Library Difference**:
> > > We used the ViT-B/16 model provided by the **Torchvision** library, while the **SAR paper** used the one from the **Timm** repository. Although both are labeled as ViT-B/16, the underlying pretrained weights are different, leading to performance variation. This discrepancy is also reflected in the **ROID paper [1]**, where the reported source-only and SAR results differ significantly from those in the original SAR paper.
> > >
> > > 2. **Number of Test Samples**:
> > > On Imagenet-C, we sampled only **5,000** test images per corruption type, whereas the SAR paper used **50,000** samples. This difference in test data size could significantly affect cumulative accuracy and convergence of the adaptation process.
> > >
> > > 3. **Learning Rate Difference**:
> > > As described in our Supplementary Material, we used the same optimizer and learning rate **(0.00025)** across all methods and batch sizes to ensure fairness. In contrast, the official SAR GitHub uses different learning rates depending on the method and batch size **( 0.000015625 for BS=2)**. Such differences can contribute to performance discrepancies.
> > >
> > > 4. **Random Seed**:
> > > Differences in the random seed can lead to slight performance variation.
> > >
> > > Taking all these factors into account, we redesigned our ViT-B/16 experiments to **perfectly match the SAR paper** settings, including using Timm, matching dataset size, learning rate, and random seed. As a result, we obtained the following **accuracy** results:
> > >
> > > | Method     |BS1 |BS2 |BS4 | BS16 | BS64 |
> > > |-------------|-----------|------------|------------|------------|------------|
> > > | Source    | 29.93 |  29.93 |  29.93 |  29.93 |  29.93 |
> > > | Tent @ LN | 47.87    | 47.89   | 47.88    |  47.94    | 48.21    |
> > > | Tent @ Buffer+LN | **48.07**    | **48.23**    | **48.25**    |  **48.57**    |**48.91**    |
> > > | | ||| | |
> > > | Eata @ LN | 46.94    | 51.91   | 55.48    |  59.62    | 61.47    |
> > > | Eata @ Buffer+LN | **48.78**    | **53.78**    | **57.09**    |  **60.12**    |**62.28**    |
> > > | | ||| | |
> > > | SAR @ LN | 56.72    | 54.81   | 56.15    |  56.41    | 56.16    |
> > > | SAR @ Buffer+LN | **61.18**    | **59.11**    | **59.48**    |  **60.22**    |**59.87**    |
> > >
> > >
> > > As shown in the table below, at batch size 1, we conducted our experiments using exactly the same settings as those reported in **Table 4 of the SAR** paper:
> > >
> > > | Method     |SAR paper | Ours |
> > > |-------------|-----------|------------|
> > > | Source |  29.9 | 29.93 |
> > > | TENT     |47.7 | 47.87 |
> > > | EATA     |46.6 | 46.94|
> > > | SAR     | 56.3 | 56.72|
> > >
> > > These results are very close to SAR paper, which validates the correctness of our implementation. Moreover, we observed that our Buffer layer continues to perform well on ViT models, further demonstrating the value of your proposed 2D reshape-based strategy.
> > >
> > > **We want to emphasize that the fact that our results changed under different settings does not imply any flaw in our original experiments. Instead, it highlights that each method must be interpreted in the context of its own experimental setup. Our Buffer module still consistently improves performance when applied in a controlled and fair setting, and this additional experiment served as a valuable double-check of our method's robustness.**
> > >
> > > We hope this addresses your concerns regarding the discrepancy with the SAR paper.
> > >
> > > Once again, we truly appreciate your comments. It took us some time to carefully design and run the experiments needed to clarify the discrepancy, and we hope our updated results provide sufficient transparency. As the discussion period has been extended by two more days, please feel free to raise any further concerns or suggestions. We will be more than happy to continue the discussion.
> > >
> > > [1] Marsden, Robert A., Mario Döbler, and Bin Yang. "Universal test-time adaptation through weight ensembling, diversity weighting, and prior correction." Proceedings of the IEEE/CVF Winter Conference on Applications of Computer Vision. 2024.

---

> > > ### Author Response · Authors · 2025-08-07
> > > **Minor Clarifications**
> > >
> > > *In our rebuttal, the ViT result above was mistakenly labeled as @BN, but it was actually @LN. We apologize for this typo.
> > >
> > > *You referred to **Table 2** of the SAR paper in your comment, but our results should be compared against **Table 4** in the SAR paper, since our experiment is not conducted under an imbalanced label shift setting.

---

> > > > ### Author Response · Authors · 2025-08-09
> > > > **Fair comparison between GN and BN**
> > > >
> > > > We have now completed the experiments on ImageNet-C using **ResNet-50 with Group Normalization (GN)** to enable a fairer comparison between BN and GN.
> > > >
> > > > | Method|      |BS1 |BS2 | BS4 | BS16 |
> > > > |-------------|-----------|------------|------------|------------|------------|
> > > > | Source|    | 69.10 | 69.10     | 69.10      |  69.10      |
> > > > | Tent |@ GN | 78.32    | 78.31    | 78.27    |  78.23    |
> > > > |  |@ Buffer | **72.20**    | **72.13**    | **72.01**    |  **70.72**    |
> > > > |  |@ GN+Buffer | 80.10    | 79.58    | 78.58    |  **77.61**    |
> > > > | | | | ||
> > > > | SAR |@ GN | 65.57    | 65.61    | 63.55  |  64.31    |
> > > > |  |@ Buffer | 68.53    | 68.77    | 68.34    |  67.60    |
> > > > |  |@ GN+Buffer | **64.72**   | **65.13**    | **59.31**    |  **63.77**    |
> > > > | | | | ||
> > > > | DeYO |@ GN | 56.52    | 57.2    | 56.81  |  53.82    |
> > > > |  |@ Buffer | 62.21    | 62.92    | 61.19    |  60.83    |
> > > > |  |@ GN+Buffer | **52.12**   | **53.10**    | **52.11**    |  **52.58**    |
> > > > | | | | ||
> > > > | ROID |@ GN | 99.91    | 88.70    | 65.92  |  64.22    |
> > > > |  |@ Buffer | 99.91    | **70.10**    | 65.56    |  63.36    |
> > > > |  |@ GN+Buffer | 99.91   | 71.82    | **64.13**    |  **58.39**    |
> > > > | | | | ||
> > > > | CMF |@ GN | 99.93   | 75.77    | 66.91  |  56.88    |
> > > > |  |@ Buffer | 99.93    | **69.38**    | **62.27**    |  60.96  |
> > > > |  |@ GN+Buffer | **99.3**   | 74.99    | 66.12    | **55.72**   |
> > > >
> > > > The reported numbers are highly comparable to those from several established TTA methods, which we believe demonstrates the fairness and appropriateness of our evaluation. Importantly, our proposed @buffer approach still yields a substantial performance improvement. This indicates that the method plays a significant role even in batch-independent settings such as GN, suggesting that it can be applied in a more general and versatile manner.
> > > >
> > > > Your review has motivated us to explore our approach under a broader range of conditions, strengthening our confidence in its generality. This constructive feedback has been invaluable in improving the quality and completeness of our work. We are sincerely grateful for your insights, which have helped us refine the paper into a higher-quality contribution.

---

### Official Review · Reviewer_SMLc · 2025-07-03

**Clarity:** 3
**Significance:** 3
**Originality:** 3
**Rating:** 4
**Confidence:** 4

**Summary:**

The paper proposes a light-weight buffer layer to address the catastrophic forgetting problem of TTA. With extensive experiments, the paper shows that this simple plug-and-play modification of the model can achieve surprisingly better performance when inserted into any other TTA methods.

**Questions:**

See the above Weakness.

**Ethical Concerns:**

["NO or VERY MINOR ethics concerns only"]

**Final Justification:**

I keep my rate since all my concerns have been addressed

**Limitations:**

yes

**Paper Formatting Concerns:**

None.

**Quality:**

3

**Strengths And Weaknesses:**

Strengths:
1. The paper is easy to read and follow. The motivation is clear.
2. The proposed module is simple and can be easily integrated into any other TTA methods, which makes it have wide applications.
3. All the experiments, including conventional CIFAR-10 & CIFAR-100C datasets, and several datasets with domain shifts, validate the effectiveness of the proposed method.

Weakness:
1. The proposed method is similar to the weight expansion of continual learning, which should be discussed in the related work.
2. How does the number of paralleled buffer layers (e.g., multiple 1x1 conv) influence the final performance?
3. In Table 1., some results of employing buffer layer drops for ROID,  which should be explained why it happens and when the proposed method works or not.
4. How does the proposed method compare to LoRA, which can be employed at BN or Conv layers?

---

> ### Author Rebuttal · Authors · 2025-07-31
>
> Thank you for your insightful comments. We appreciate the opportunity to improve our manuscript and have addressed your points below.
>
> ---
>
> W1 :  **Weight expansion of continual learning**
>
> R1: Although our Buffer Layer is applied only at test time—whereas classic weight‑expansion methods add modules during training—both paradigms freeze the backbone parameters and introduce compact auxiliary layers to mitigate catastrophic forgetting. Recognizing this conceptual similarity, we will expand the Related Work section with a comprehensive comparison to these training‑time expansions (e.g., Progressive Neural Networks, adapter modules) and recent low‑rank adaptation techniques such as LoRA.
>
>
> ---
>
> W2 : **The number of paralleled buffer layers** (e.g., multiple 1x1 conv)
>
> R2 : Thank you for this excellent question. In the main manuscript (Fig. 4 & Table 5) we focused on the type of Buffer layer, but examining how the number of parallel layers affects performance is indeed very insightful. As shown below, we swept from 1 up to 5 parallel 1×1 buffers (and one mixed 1×1 + 3×3 + 5×5) under batch sizes 4 and 16 on CIFAR10-C with WRN28 architecture, as with our main manuscript :
>
> -----
>
> | Configuration    | BS = 4 Error (%) | BS = 16 Error (%) |
> |------------------|------------------|-------------------|
> | single 1×1conv         | 29.16            | 21.22             |
> | double 1×1conv          | 29.02            | 20.87             |
> | 1×1conv + 3×3conv        | 29.15            | **20.37**             |
> | 3* 1×1conv           | 29.07            | 20.71             |
> | 1×1conv + 3×3conv + 5×5conv  | 29.83            | 20.41             |
> | 4* 1×1conv           | **28.96**            | 20.58             |
> | 5* 1×1conv          | **28.96**            | 20.56             |
>
> As shown in the table above, an intriguing phenomenon emerges as we increase the number of parallel 1×1 conv buffers: under very small batch sizes (BS = 4), error steadily decreases—from 29.16% with a single 1×1 layer down to 28.96% with four or more—demonstrating the clear benefit of extra 1×1 paths in small‑batch regimes. However, these pure 1×1 expansions remain suboptimal at BS = 16, never outperforming our original 1×1 + 3×3 configuration. Conversely, the 1×1 + 3×3 + 5×5 mixed buffer matches BS = 16 performance but falters at BS = 4, highlighting a distinct trade‑off between kernel complexity and batch size. Overall, across both small and moderate batches, our 1×1 + 3×3 design proves the most robust and reliable choice. We appreciate this insightful question—further exploration of buffer count and kernel interactions is indeed a promising direction for future work.
>
> ---
>
> W3 : **Performance drop for ROID**
>
> R3 : We thank the reviewer for this insightful point. We hypothesize that ROID’s weight ensembling—implemented via an exponential moving average over the buffer’s randomly initialized weights—biases the buffer back toward its noisy starting state, impeding effective adaptation. To test this, we removed the weight-ensembling component and applied our Buffer Layer. The CIFAR-10-C error rates are reported below:
>
> | ROID | BS 2 | BS 4 | BS 16 |
> |------------------|------------------|-------------------|-------------------|
> | @BN        | 93.22            | 51.95             | 34.01 |
> | @Buffer         | 72.92           | 50.67             | 35.15|
> | @Buffer, no EMA        | **71.16**            | **47.73**        | **33.86**|
>
> Removing weight ensembling consistently reduces error across all batch sizes, confirming that the ensembling bias toward random initial weights was the primary bottleneck. We will include these ablation results and their analysis in our revised manuscript.
>
> ---
>
> W4 : **Comparison to LoRA**
>
> R4 :For a fair comparison, we built a LoRA-based test-time adaptation variant on WideResNet-28 (WRN28) using the official ConvLoRA implementation. Additionally, we inserted LoRA adapters into the same early convolutional blocks where our Buffer Layer is placed, and updated only those adapters with TENT’s entropy-minimization objective at test time. The resulting CIFAR-10-C error rates are shown below:
>
> | Method      | Config     | BS2 |BS4 | BS16 |
> |-------------|------------|------------|-------------|--------------|
> | Tent| LoRA(early)    | 39.20       | 29.85       | 21.02        |
> | |  LoRA(full)       | 87.13      | 66.42       | 24.39        |
> |         | @BN         | 82.56      | 51.36       | 23.12        |
> |             | @Buffer     | **37.05**  | **29.11**       | **20.32**    |
>
> Our Buffer Layer consistently outperforms the low-rank weight modulation of LoRA under the same test-time adaptation regime. We believe this is because domain shifts often manifest as localized, spatially varying distortions that are better addressed by our convolutional residual branches, whereas global low-rank updates struggle to correct such input-specific feature misalignments.
>
> Thank you for this insightful question. Addressing it has helped us deepen our analysis and strengthen our empirical results. We will add this comparison and discussion to the revised manuscript. Please let us know if you have any further comments!

---

> > ### Author Response · Authors · 2025-08-06
> >
> > We thank the reviewer for taking the time to read our rebuttal and for their thoughtful initial feedback. We look forward to any further comments the reviewer may have and would be grateful for any additional suggestions or concerns. We remain open and eager to address them in order to further improve our work.

---

> > ### Comment · Reviewer_SMLc · 2025-08-09
> > **Thanks for the rebuttal**
> >
> > The authors have addressed all my concerns, and  I will keep my rate.

---

### Note · Authors · 2025-08-13

We sincerely thank all reviewers for their constructive feedback and thoughtful questions throughout the review and discussion phases. These interactions led us to explore new aspects, verify our claims, and broaden our analysis. Below, we summarize the key points clarified and strengthened during this process.

**Generality to Different Architectures** : Several reviewers asked about the applicability of our method to non-CNN architectures, particularly Transformers. Motivated by these comments, we applied the same logic from our CNN experiments(parallel, randomly initialized convolutional branches) by reshaping Transformer tokens into 2D and attaching the Buffer Layer directly. This yielded consistent, meaningful gains, confirming that our approach is not restricted to CNNs but extends to a broader range of architectures. This represents a promising direction for future work.

**Comparison to PEFT Methods** : Multiple reviewers noted similarities to PEFT approaches such as LoRA. While they share some conceptual similarities, PEFT is designed for supervised fine-tuning with labeled data, whereas ours is unsupervised TTA. During the discussion phase, we empirically confirmed that the proposed Buffer Layer consistently outperforms LoRA-based variants under identical TTA conditions.


**Fairness of Reported Results** : There were raised concerns about discrepancies with the numbers reported in a specific prior work. We carefully reproduced those settings (matching architectures, datasets, learning rates, and seeds) and demonstrated that the differences were due to variations in experimental setup. Across all experiments, including those added during the discussion, we maintained fair and consistent evaluation protocols.

**Additional Analyses** : The discussion encouraged us to conduct additional analyses on why the Buffer Layer performs well in some scenarios but less effectively in others, as well as to examine forgetting behavior and assess computational trade-offs, enabling us to clarify and strengthen our discussion in these aspects. We also provided fair, reproducible comparisons across normalization schemes, architectures, and prior methods.

We appreciate the opportunity to address these points and believe the manuscript would have improved clarity, scope, and empirical support. We trust that our thorough responses, particularly those provided during the discussion phase, will be fully considered in the final evaluation.

---

### Decision · Program_Chairs · 2025-09-17

**Decision:**

Accept (poster)

**Comment:**

This paper introduces a Buffer Layer for test-time adaptation (TTA), a lightweight parallel module that adapts at inference while keeping the backbone frozen. The design addresses the instability of normalization-based TTA and mitigates catastrophic forgetting. The method is simple, modular, and effective, showing consistent gains across diverse datasets and TTA baselines. The authors provided extensive additional experiments (e.g., ViTs, GN backbones, LoRA comparisons, initialization and complexity analyses) that strengthened claims of robustness and generality. Conceptually related to PEFT methods, with somewhat incremental novelty. Some evaluation details (e.g., SAR results, computational cost, Transformer applicability) remain limited, though the rebuttal addressed them substantially. Despite minor concerns, the paper makes a solid and practical contribution to TTA. I recommend acceptance (poster).